# FreqPolicy: Efficient Flow-based Visuomotor Policy via Frequency Consistency

**Yifei Su**[1,2*]**, Ning Liu**[1*†]**, Dong Chen**[1]**, Zhen Zhao**[1]**, Kun Wu**[1]
**Meng Li**[1]**, Zhiyuan Xu**[1]**, Zhengping Che**[1†‡]**, Jian Tang**[1‡]
[1]Beijing Innovation Center of Humanoid Robotics
[2]NLPR, MAIS, Institute of Automation of Chinese Academy of Sciences

## Abstract

Generative modeling-based visuomotor policies have been widely adopted in robotic manipulation, attributed to their ability to model multimodal action distributions. However, the high inference cost of multi-step sampling limits its applicability in real-time robotic systems. Existing approaches accelerate sampling in generative modeling-based visuomotor policies by adapting techniques originally developed to speed up image generation. However, a major distinction exists: image generation typically produces independent samples without temporal dependencies, while robotic manipulation requires generating action trajectories with continuity and temporal coherence. To this end, we propose FreqPolicy, a novel approach that first imposes frequency consistency constraints on flow-based visuomotor policies. Our work enables the action model to capture temporal structure effectively while supporting efficient, high-quality one-step action generation. Concretely, we introduce a frequency consistency constraint objective that enforces alignment of frequency-domain action features across different timesteps along the flow, thereby promoting convergence of one-step action generation toward the target distribution. In addition, we design an adaptive consistency loss to capture structural temporal variations inherent in robotic manipulation tasks. We assess FreqPolicy on $53$ tasks across $3$ simulation benchmarks, proving its superiority over existing one-step action generators. We further integrate FreqPolicy into the vision-language-action (VLA) model and achieve acceleration without performance degradation on $40$ tasks of Libero. Besides, we show efficiency and effectiveness in real-world robotic scenarios with an inference frequency of $93.5$ Hz.

## 1 Introduction

Recently, the generative modeling-based visuomotor policy framework extends the powerful capabilities of generative models in text-to-image synthesis [58, 17, 56, 60, 9, 21] to imitation learning for robotic manipulation [27, 11, 39, 19, 6], achieving significant breakthroughs. A prominent class of such policies is diffusion-based approaches [11, 76, 46, 75, 83, 23], which are widely adopted for the ability to model complex multimodal distributions in high-dimensional action spaces [11]. More recently, flow matching [40, 38, 13, 36] has emerged as a generalization of diffusion models, offering simpler optimization objectives and more stable training [68]. It has been applied to robotics [59, 19, 12, 77, 6, 25], proving the efficacy of flow-based policies. Despite these advancements, these generative modeling-based visuomotor policies rely on an iterative sampling process to transform Gaussian noise into actions, resulting in high-latency inference. This presents a significant

---

[*]Co-first authors: Yifei Su and Ning Liu.

[†]Project leaders: Ning Liu and and Zhengping Che.

[‡]Corresponding authors: Jian Tang and Zhengping Che.

39th Conference on Neural Information Processing Systems (NeurIPS 2025).

bottleneck for real-time inference in robotic manipulation, where stable and smooth action execution is essential, particularly in long-horizon or dynamic tasks [53, 42].

To mitigate this issue, recent efforts leverage acceleration methods from the image generation domain to accelerate the action generation by reducing the sampling steps [53, 42, 28, 65, 16]. A pioneering work, Consistency Policy (CP) [53], adapts the consistency distillation from the text-to-image domain [62, 31, 43, 64, 41] to the robotics domain. CP first trains a powerful diffusion-based teacher using the EDM model [29], and subsequently distills its knowledge into a student model via the Consistency Trajectory Model objective [31], enabling the diffusion-based policy one-step action generation. ManiCM [42] extends the consistency distillation to 3D diffusion-based policy [76] with point cloud inputs. SDM [28] and OneDP [65] adopt distribution matching distillation originally introduced in image generation [52, 66, 71, 14], distilling pretrained diffusion-based policies into one-step action generators by aligning the action distributions. While the aforementioned methods primarily focus on diffusion-based policies, FlowPolicy [78] attempts to reduce the sampling step of flow-based policy. It uses consistent constraints from Consistency-FM [68] originally developed for image generation, to enforce velocity consistency across different points along the flow without distillation, facilitating straight flows and enabling one-step generation.

In fact, a key distinction between image generation and robotic manipulation lies in the presence of temporal dependencies. While image generation produces individual samples that are typically independent across time, robotic manipulation involves executing action trajectories that form a time-series process. Therefore, it requires modeling outputs that are continuous and temporally coherent. We argue that *leveraging temporal characteristics is essential, as it provides richer contextual information that can significantly enhance action generation.* In this work, we proposed a novel one-step visuomotor policy, named FreqPolicy, that first imposes frequency consistency constraints on flow-based visuomotor policy, thus leveraging temporal knowledge and enabling efficient and high-quality one-step action generation. Specifically, we introduce a frequency consistency objective that enforces alignment of frequency-domain action features across timesteps, improving temporal coherence in generated actions. Furthermore, we design an adaptive frequency component loss to learn the structural temporal variations inherent in robotic manipulation tasks.

**Frequency Consistency Objective.** Inspired by the fields of time-series forecasting and speech processing [82, 81, 70, 73], frequency representations capture better non-stationary and oscillatory patterns, where time-domain features often fail. Thus, frequency-domain features show high effectiveness for modeling temporal dynamics. In robotic manipulation, for high-frequency sampled action chunks, frequency features provide finer discrimination of subtle variations in smooth trajectories. To exploit this advantage, we enforce consistency between velocities of action chunks across different timesteps in the frequency space, thereby promoting the convergence of action generation.

**Adaptive Frequency Component Loss.** In robotic manipulation, the distribution of frequency components within each action chunk varies over time throughout task execution. This variability arises because robotic manipulation sequences typically alternate between stationary and non-stationary motion phases. During low-dynamic movements (e.g., reaching or moving between positions), only a subset of action dimensions exhibit noticeable variation, while others remain relatively smooth. Conversely, during high-dynamic movements (e.g., transitions between skills or contact-rich interactions), high-frequency variations are more prominent and informative. To capture this structured temporal variation, we draw inspiration from the Focal Loss [35] and propose an adaptive weighting scheme that dynamically emphasizes frequency components with greater discrepancy.

Our contributions can be summarized as: 1) FreqPolicy is the first one-step visuomotor policy that imposes temporal knowledge for robotic manipulation. 2) Inspired by the time-series and speech processing field, we propose the frequency consistency constraint objective to enhance the regularization of two arbitrary action velocities. Moreover, an adaptive frequency component loss is proposed to capture the structured temporal variation of the action sequence. 3) We conduct extensive experiments in both simulation and the real world to evaluate FreqPolicy, demonstrating its superiority over existing one-step action generators, e.g., achieving $78.5\%$ in MetaWorld. 4) We further integrate FreqPolicy into a vision-language-action (VLA) model [6, 32], achieving a significant improvement in inference speed (e.g., $5\times$ faster) without compromising overall task performance.

## 2 Related Work

**Generative Modeling-based Visuomotor Policy.** Learning from human demonstrations (a.k.a. imitation learning [2, 51, 20, 1]) has shown remarkable performance in robotic manipulation [8, 7, 34, 33]. Compared to deterministic policies [50, 79, 26, 80, 4, 18, 54], generative modeling-based policies are widely used for their ability to model multimodal action distributions, enhancing training stability in high-dimensional action spaces [11]. The pioneer work, Diffusion Policy [11], formulated visuomotor policy learning as a conditional denoising diffusion process, achieving impressive manipulation performance through a multi-step sampling inference process. Subsequent works [76, 30] extended diffusion-based policies to 3D point cloud inputs, enabling action generation directly from point cloud observations. [46, 67] factorizes the policy into a high-level key waypoint predictor and low-level trajectory generator, achieving higher task success rates. Recently, another generative modeling paradigm, Flow Matching (FM) [40, 38] has shown strong performance in domains including image generation [13, 45, 17] and super-resolution [84]. Compared to diffusion-based generative models, flow matching (FM) directly defines probability paths via ordinary differential equations (ODEs) to transport the simple prior distribution to a target distribution, offering better numerical stability and fewer inference steps [36]. Hence, it has been applied to robotic manipulation [6, 25, 61, 5]. As a pioneer, Rouxel *et al.* [59] extend flow matching to multi-support tasks and demonstrated strong performance. Subsequent works [19, 12] incorporate point cloud inputs, achieving manipulation in 3D scenery. Zhang *et al.* [77] innovatively integrate spatial affordance prediction with flow matching for action prediction.

**Accelerated Visuomotor Policy.** Despite the impressive performance of generative-based policies, they are limited in real-world deployment by the high inference cost of multi-step sampling [53]. To address this issue, recent efforts have accelerated action sampling through four main mainstream as: (1) *Trajectory distillation.* Consistency Policy [53] pioneered consistency distillation [62, 43, 31] in robot manipulation, enabling a few-step action generation by constraining denoising trajectories from different steps toward the same step. ManiCM [42] extended this approach to 3D scenarios and achieved better one-step inference than 3D Diffusion Policy [76]. FlowPolicy [78], inspired by ConsistencyFM [68], applied constraints on the velocity field and transporting process in flow matching, enabling one-step action generation. (2) *Partial action denoising.* Instead of denoising from Gaussian noise per step, SDP [22] outputs a partially denoised action with variable levels of noise, where the noise-free part guides current action execution and the noisy portion serves as input for future rollouts. Falcon [10] proposed to select an action chunk from the historical denoised trajectory as the input. Both enhance the inference speed but still require multiple sampling steps. (3) *Distribution matching.* OneDP [65] and SDM [28] incorporated variational score distillation [66, 44, 72, 71, 48, 14] into imitation learning and achieved strong one-step generation performance in both 2D [47] and 3D [74] input settings. Although effective, such approaches typically require a pretrained teacher model. In addition to the above methods, some works have introduced advanced Riemannian flow matching [15] and variance-based adaptive sampling strategies [24] to achieve faster generation, respectively. This work aims to accelerate flow-based visuomotor policies using consistency-based paradigms, motivated by their potential to eliminate the need for teacher models. Unlike existing works, we propose a novel perspective based on frequency consistency to enable more effective one-step action generation by leveraging temporal knowledge in robotic manipulation.

## 3 Methodology

### 3.1 Task Setting

We focus on the robotic manipulation task via the imitation learning paradigm: given a dataset $\mathcal{D}$ containing $n$ observations $\mathcal{O} = \{o\}_{i=1}^{n}$ and corresponding expert action $\mathcal{A} = \{a\}_{i=1}^{n}$, the goal is to train a flow matching-based policy $\pi_\theta(a_{1:H}|o) : \mathcal{O} \rightarrow \mathcal{A}$ to map the observations $o \in \mathcal{O}$ to action $a_{1:H}$ with chunking size $H$, enabling the robot to replicate expert behaviors and generalize across diverse scenarios. Observations typically include proprioceptive states, RGB images from various viewpoints (e.g., eye-in-hand or third-person cameras), and point cloud data. The action space varies depending on the task and robot setup, and commonly includes either SE(3) motions of the end-effector or joint states. To thoroughly evaluate the effectiveness of the proposed method, we conduct extensive experiments under both 2D and 3D observation settings.

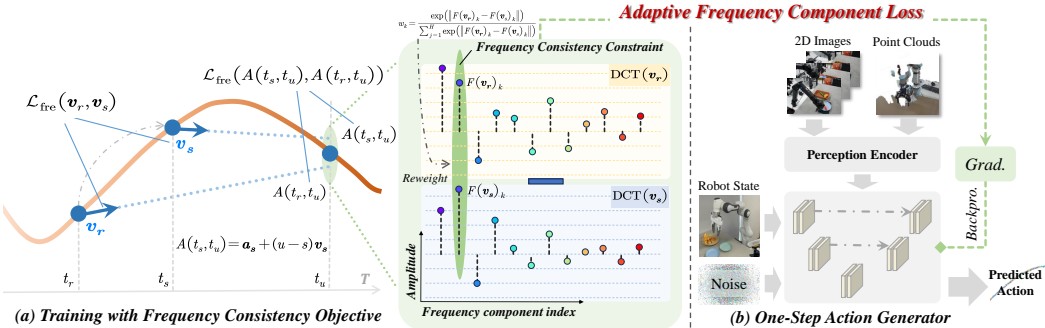

*(a) Training with Frequency Consistency Objective*    *(b) One-Step Action Generator*

Figure 1: **Overview of FreqPolicy.** a) For training, we use the frequency consistency constraint to align the velocity vectors across different time steps in the frequency space. Besides, we introduce an adaptive frequency component loss to accommodate the diverse frequency structures in manipulation tasks. b) FreqPolicy takes 2D or 3D input and predicts the velocity vector of the action as output.

## 3.2 Preliminaries

**Flow Matching.** Let $\mathcal{R}^d$ denote the data space with data points $x \in \mathcal{R}^d$, and consider a simple prior distribution $x_0 \sim p_0(x)$, FM aims to learn a time-dependent vector field $v(t, x) : [0, 1] \times \mathcal{R}^d \to \mathcal{R}^d$, that generates a time-conditioned ordinary differential equation (ODE) as follows:

$$\frac{d\psi(t, x)}{dt} = v(t, \psi(t, x)), \ \ \psi(0, x) \sim p_0 \tag{1}$$

$\psi(\cdot) : [0, 1] \times \mathcal{R}^d \to \mathcal{R}^d$ is the solution to the ODE, also known as the *flow*. Based on this, the distribution $p_0$ can be transported to a more complex distribution $p_t$ via the push-forward equation:

$$p_t = p_0 \left( \psi^{-1}(t, x) \right) \det \left[ \frac{\partial}{\partial x} \psi^{-1}(t, x) \right] \tag{2}$$

To transform the prior distribution $p_0$ into an unknown target distribution $p_1$ via flow $\psi$, with the marginal distribution satisfying $p_{t=0} = p_0$ and $p_{t=1} = p_1$, flow matching seeks to optimize the vector field $v(t, x)$ by minimizing the following objective:

$$\mathcal{L}_\theta = \mathbb{E}_{t, p_t} \left\| v_\theta(t, x_t) - u(t, x_t) \right\|_2 \tag{3}$$

where $\theta$ denotes the learnable parameters of the vector field, which in turn leads to a deep parametric model of the flow $\psi$. $u(t, x_t)$ is the corresponding target vector field. Despite the simplicity of the objective, directly optimizing it in practice is challenging due to the lack of prior knowledge of the desired $p_t$ and $u(t, x_t)$. Therefore, Conditional Flow Matching [36] proposes to regress $v_\theta(t, x_t)$ via a conditional vector field $u(t, x_t | x_1)$ and a conditional probability path $p_t(x_t | x_1)$ as:

$$\mathcal{L}_\theta = \mathbb{E}_{t, p_1(x_1), p_t(x_t | x_1)} \left\| v_\theta(t, x_t) - u(t, x_t | x_1) \right\|_2 \tag{4}$$

Here Eq. 3 and Eq. 4 share the same gradient with respect to $\theta$, thus Eq. 3 can be estimated if $u(t, x_t | x_1)$ and $p(t, x_t | x_1)$ is tractable. The choice of $u(t, x_t | x_1)$ is not unique, and a classic instance is obtained by linearly interpolating between $x_t$ and $x_1$ [40, 38], which owns the desirable property of generating straight probability paths:

$$u(t, x_t | x_1) = \frac{x_1 - x_t}{1 - t} \tag{5}$$

$$\mathcal{L}_\theta = \mathbb{E}_{t, p_1(x_1), p_t(x_t)} \left\| v_\theta(t, \psi(t, x_t)) - (x_1 - x_t) \right\|_2 \tag{6}$$

## 3.3 FreqPolicy

Flow-based policies suffer from slow inference due to their iterative sampling process, limiting their use in high-frequency control scenarios and deployment on edge devices. We propose FreqPolicy, a novel approach that enables efficient one-step action generation by introducing temporal constraints from the frequency perspective, as shown in Fig. 1. Inspired by the time-series and speech processing fields, we propose the frequency consistency constraint objective to enhance the regularization of two

arbitrary action velocities. Besides, the adaptive frequency component loss is proposed to effectively capture the structured temporal variation of the action sequence. We provide details on the model architecture in Sec. A.1.

**Basic Flow Matching Learning Objective.** An intuitive way to apply flow matching in robotic manipulation is to regress a vector field mapping noise to action chunks, as adopted in prior works [19, 59, 12, 24, 6]. We first equip FreqPolicy with this objective to enable multimodal action generation.

Specifically, considering a simple noise $a_0 \sim \mathcal{N}(0, I)$ and a corresponding expert action $a_1 \sim p_{\text{expert}}$, we aim to learn the vector field $v_\theta(t, a_t)$ that maps the prior noise to expert actions by approximating the real conditional velocity field $u(t, a_t)$. Following flow matching based on optimal transport theory [40, 38], we instantiate the target velocity field $u(t, a_t)$ as a constant vector along the optimal linear transport from $a_0$ to $a_1$, namely:

$$u(t, a_t) = \int_0^1 u(z, a_z)dz = a_1 - a_0 \tag{7}$$

Next, for any timestep $t \in [0, 1]$, we optimize the policy by minimizing the following objective:

$$\mathcal{L}_{\text{fm}} = \mathbb{E}_{t \sim \mathcal{U}(0,1), \, (a_0, a_1) \sim \mathcal{D}} \left\| v_\theta(t, a_t) - (a_1 - a_0) \right\|_2 \tag{8}$$

where $a_t$ is defined as the linear interpolation between $a_0$ and $a_1$ with respect to time $t$, $a_t = (1 - t) \cdot a_0 + t \cdot a_1$. For inference, we sample an initial noise vector $a_0$ and deterministically integrate the learned vector field $v_\theta(t, a_t)$ from $t = 0$ to $t = 1$ using an ODE solver (e.g., Euler or Runge-Kutta), yielding the final action $a_1$. While optimizing the objective in Eq. 8 yields a functional policy model, it still requires multiple sampling steps to generate a desired action (e.g., 10 steps in $\pi_0$ [6] and 4 steps in GR00T N1 [5]).

**Frequency Consistency Constraint Objective.** In the fields of time-series analysis and speech processing [82, 81, 70, 73], frequency-domain features show superior ability to model temporal dynamics, e.g., non-stationary [69] and oscillatory patterns, where time-domain signals often fall short. In robotic manipulation, frequency features provide finer discrimination of subtle variations in high-frequency sampled action chunks [49]. Therefore, differing from prior approaches [78] that treat multi-dimensional action chunks as static vectors, the proposed FreqPolicy regards each action chunk as a temporal signal. We aim to enforce consistency in the frequency space between action velocity vectors across different timesteps of the flow, promoting straighter flow and one-step action generation. Concretely, given a pair of initial noise and target actions $(a_0, a_1)$ sampled from the prior and expert distributions, we select two arbitrary timesteps $s, r \in [0, 1]$, and then construct the interpolated state between $a_0$ and $a_1$ as:

$$a_r = (1 - r) \cdot a_0 + r \cdot a_1, \quad a_s = (1 - s) \cdot a_0 + s \cdot a_1 \tag{9}$$

We enforce the velocities $v_\theta(s, a_s)$ and $v_\theta(r, a_r)$ to be consistent across timesteps through the following training objectives:

$$\mathcal{L}_{\text{freq}} = \mathbb{E}_{r,s \sim \mathcal{U}(0,1), \, (a_r, a_s) \sim \mathcal{D}} \left[ \text{Sim}(v_\theta(s, a_s), v_\theta(r, a_r)) \right] + \\ \mathbb{E}_{r,s,u \sim \mathcal{U}(0,1), \, (a_r, a_s, a_u) \sim \mathcal{D}} \left[ \text{Sim}(a_s + (u - s) \, v_\theta(s, a_s), a_r + (u - r) \, v_\theta(r, a_r)) \right] \tag{10}$$

where $\text{Sim}(\cdot)$ is a function measuring the consistency of two velocities, and $u$ satisfies $r < s < u$. The first part of Eq. 10 directly ensures the consistency of the two velocity vectors, while the second part enforces the consistency of the vector field from a trajectory perspective. Specifically, starting from arbitrary two points, $a_r$ and $a_s$, are expected to converge to the same point at time $u$, thereby providing a way to directly define straight flows [68]. These constraints enforce consistent and straight flow across different timesteps, thus facilitating effective one-step action generation.

To leverage the temporal characteristic of the action chunks, we strengthen the temporal constraint with the frequency regularization. Namely, we propose to project the velocity vectors into the frequency domain using the type-II Discrete Cosine Transform (DCT):

$$F(v_t)_k = \sum_{n=0}^{H-1} v_t(n) \cdot \cos\left[ \frac{\pi}{N} \left( n + \frac{1}{2} \right) k \right], \quad \text{for } k = 0, \ldots, H - 1 \tag{11}$$

where $F(v_t)_k$ denotes the spectral coefficient of the $k$-th frequency component of $v_\theta(t, a_t)$. The function $\text{Sim}(\cdot)$ in Eq. 10 is then defined as the $\ell_2$ norm between the two frequency coefficients as:

$$\text{Sim}(v_r, v_s) = \left\| F(v_r) - F(v_s) \right\|_2 \tag{12}$$

This loss encourages the frequency spectral profiles of velocity signals at different timesteps to match, thereby aligning their temporal dynamics. As a result, the action chunks generated across different timesteps exhibit greater temporal consistency, which in turn guides the policy towards learning straighter and more stable flow trajectories suitable for one-step inference.

**Adaptive Frequency Component Loss.** In robotic manipulation, the distribution of frequency components within each action chunk varies over time throughout task execution. This variability arises because robotic manipulation sequences typically alternate between stationary and non-stationary motion phases. During low-dynamic movements (e.g., reaching or moving between positions), only a subset of dimensions in the action chunks exhibit noticeable variation, while others remain relatively smooth. Conversely, during high-dynamic movements (e.g., transitions between skills or contact-rich interactions), high-frequency variations are more prominent and informative. We show some visualization examples in Sec. A.2. To effectively capture this structured temporal variation, we draw inspiration from the Focal Loss [35] and propose an adaptive weighting scheme that emphasizes frequency components with greater discrepancy. Concretely, for each frequency band $k$, we compute a frequency-domain weight $w_k$ based on the difference between the corresponding frequency components of two action velocity vectors:

$$w_k = \frac{\exp\left(\|F(v_r)_k - F(v_s)_k\|_2\right)}{\sum_{j=0}^{H-1} \exp\left(\|F(v_r)_j - F(v_s)_j\|_2\right)} \tag{13}$$

where $F(v_t)_k$ is the DCT coefficient of the $k$-th frequency component. We then define the adaptive frequency component loss as:

$$\text{Sim}(v_r, v_s) = \sum_{k=0}^{H-1} w_k \cdot \|F(v_r)_k - F(v_s)_k\|_2 \tag{14}$$

This adaptive loss encourages the model to focus more on frequency bands that encode informative dynamic variation across time. As a result, it strengthens the model's ability to align the temporal structure of action chunks across different timesteps. We investigate the effectiveness of the adaptivity capability of the proposed loss in Sec. 4.3.

**Overall Training Objective.** Our training objective consists of two complementary components: a standard flow matching loss in Eq. 8 supervising the flow from prior noise to expert actions, and a frequency-domain loss in Eq. 14 that enforces temporal consistency across various timesteps. We jointly optimize these two objectives through the final loss:

$$\mathcal{L}_{\text{total}} = \mathcal{L}_{\text{fm}} + \mathcal{L}_{\text{freq}} \tag{15}$$

This unified formulation ensures that FreqPolicy learns both accurate and temporally consistent flows, ultimately enabling reliable one-step action generation across diverse manipulation tasks.

## 4 Experimental Results

### 4.1 Simulation Experiments

To thoroughly validate the FreqPolicy, we conduct simulation experiments under both 2D [47] and 3D input settings [74, 3, 55] as shown in Fig. 2, and perform comprehensive comparisons. We further integrate FreqPolicy with existing vision-language-action (VLA) models to demonstrate its generalization ability.

**Experiment with 2D Inputs.** The FreqPolicy is evaluated on 5 tasks from the widely-used Robomimic [47] benchmark: Lift, Can, Square, Transport, and Tool Hang. For each task, we use proficient human demonstrations datasets with image-based observations, containing 200 demonstrations per task. We evaluate the average success rate across 3 random seeds for all tasks. For each seed, we evaluate each task over 50 different initializations and compute the mean success rate. The Number of Function Evaluations (NFE) metric is reported to measure the number of generation steps.

Table 1 compares our approach with existing policies on the Robomimic benchmark. FreqPolicy surpasses prior one-step methods, e.g., CP [53] and IMLE Policy [57]. Moreover, FreqPolicy even outperforms a few classical multi-step policies. We further reproduced Consistency-FM [68] that uses spatial-domain constraints for robotic manipulation with the same setting. FreqPolicy

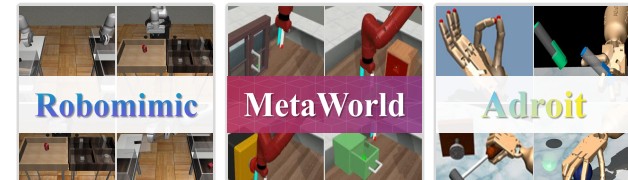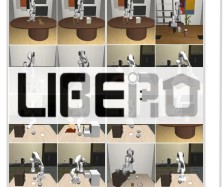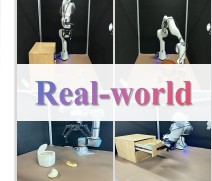

Figure 2: **Experimental benchmarks.** We evaluate FreqPolicy on 5 benchmarks, including a total of 93 simulated tasks (*left*) and 3 real robotics tasks (*right*).

Table 1: **Performance on the Robomimic dataset** [47]. We compare the FreqPolicy against a set of state-of-the-art multi-step generative visuomotor policies. We report the mean and standard deviation of success rates. Tasks marked with $*$ are the result of our implementation.

| Method | NFE | Lift | Can | Square | Transport | Toolhang |
|---|---|---|---|---|---|---|
| DDPM [11] | 15 | 1.00 | $0.98 \pm .01$ | $0.91 \pm .01$ | $0.80 \pm .04$ | $0.52 \pm .05$ |
| DDiM [11] | 15 | 1.00 | $0.99 \pm .01$ | $0.92 \pm .03$ | $0.79 \pm .04$ | $0.55 \pm .05$ |
| RectifiedFlow* [40] | 15 | 1.00 | $0.96 \pm .02$ | $0.90 \pm .02$ | $0.84 \pm .04$ | $0.90 \pm .02$ |
| ActionFlow [19] | 10 | 1.00 | $0.99 \pm .01$ | $0.87 \pm .10$ | - | $0.81 \pm .09$ |
| AdaFlow [24] | - | 1.00 | 1.00 | 0.98 | 0.92 | 0.88 |
| ConsistencyPolicy* [53] | 3 | 1.00 | $0.95 \pm .02$ | $0.96 \pm .01$ | $0.88 \pm .02$ | $0.77 \pm .03$ |
| DDiM [11] | 1 | 0.04 | $0.00 \pm .00$ | $0.00 \pm .00$ | $0.00 \pm .00$ | $0.00 \pm .00$ |
| ConsistencyPolicy* [53] | 1 | 1.00 | $0.98 \pm .01$ | $0.92 \pm .02$ | $0.78 \pm .03$ | $0.70 \pm .03$ |
| Consistent-FM* [68] | 1 | 1.00 | $0.94 \pm .02$ | $0.90 \pm .01$ | $0.84 \pm .02$ | $0.80 \pm .02$ |
| IMLE Policy [57] | 1 | 1.00 | 0.98 | 0.82 | 0.90 | 0.81 |
| **Ours** | 1 | 1.00 | $\mathbf{0.98 \pm .02}$ | $\mathbf{0.92 \pm .02}$ | $\mathbf{0.90 \pm .02}$ | $\mathbf{0.85 \pm .03}$ |

achieves superior performance, improving the Transport task by $6\%$ and the Tool Hang task by $5\%$, demonstrating the effectiveness of the proposed frequency-domain consistency.

**Experiment with 3D Inputs.** To implement fair comparisons, we follow the existing methods [76, 28] and conduct experiments on 53 tasks across two benchmarks: Adroit [55] and MetaWorld [74]. Specifically, we use reinforcement learning with VRL3 [63] to collect expert demonstrations for Adroit, and use scripted policies to obtain demonstrations for MetaWorld. Training is conducted using 10 expert demonstrations per task. Following the evaluation protocol in previous works [76, 28, 78], we report the performance for each task across 3 random seeds. For each seed, we evaluate 20 segments every 200 training epochs and compute the average success rate of the top-5 trails, along with the average inference time per task.

Table 2 presents the comparisons with previous methods and demonstrates consistently improved performance. Compared to the SDM [28] strategy using distribution matching distillation, FreqPolicy achieves consistent gains in average success rate across 3 benchmarks, e.g., MetaWorld rises from $74.8\%$ to $78.5\%$. Unlike SDM, FreqPolicy eliminates the dependence on pretrained teachers. Moreover, against FlowPolicy [78] applying a direct consistency constraint in the spatial domain, FreqPolicy maintains a $1.3\%$ lead on MetaWorld. This advantage is particularly pronounced on the Medium, Hard, and Very Hard splits of MetaWorld, further validating the benefit of our frequency consistency supervision. Please refer to Sec. A.3 for details.

**Experiment with VLA settings.** We combine different policies with the classic OpenVLA model [33] and conduct systematic experiments on the LIBERO simulation benchmark [37], covering 4 task suites: LIBERO-Spatial, LIBERO-Object, LIBERO-Goal, and LIBERO-Long. Each suite provides 500 expert demonstrations across 10 tasks, designed to evaluate policy generalization across varying spatial layouts, object types, goal specifications, and long-horizon tasks. All models are evaluated under the same experimental protocol, with results averaged success rate (%) over 500 trials per suite (10 tasks $\times$ 50 episodes). Please refer to Sec. A.4 for more details.

We integrate FreqPolicy, Diffusion Policy (DP) [11], and Flow Matching (FM) [59, 19] with the OpenVLA. Table 3 shows the comparisons of VLA models with various policy heads. Our method (OpenVLA-FreqPolicy) achieves the highest average success rate of $94.8\%$, outperforming both

Table 2: **Performance on the benchmarks with 3D inputs** [74, 55]. We assess performance on 53 challenging tasks with 3 random seeds, reporting the mean success rate (%) and standard deviation. Tasks marked with ∗ are reproduced results.

| Method | NFE | Adroit (3) | Meteworld Easy (28) | Metaworld Medium (11) | Metaworld Hard (6) | Metaworld Very Hard (5) | Average |
|---|---|---|---|---|---|---|---|
| DP [11] | 10 | 31.7 | 83.6 | 31.1 | 9.0 | 26.6 | $55.5 \pm 3.6$ |
| DP3∗ [76] | 10 | **74.3** | 89.0 | **72.7** | 38.0 | 75.8 | $76.1 \pm 2.3$ |
| ManiCM [42] | 1 | 72.3 | 83.6 | 55.6 | 33.3 | 67.0 | $69.0 \pm 4.6$ |
| SDM [28] | 1 | 74.0 | 86.5 | 65.8 | 35.8 | 71.6 | $74.8 \pm 4.5$ |
| FlowPolicy∗ [78] | 1 | 69.3 | 89.6 | 66.5 | 43.8 | 75.8 | $77.2 \pm 2.8$ |
| **Ours** | 1 | 72.6 | **90.6** | 65.8 | **46.8** | **79.4** | $\mathbf{78.5 \pm 2.6}$ |

Table 3: **Success rates and inference speed on LIBERO benchmark** [37]. Methods with ∗ indicate our implementations. FreqPolicy (NFE = 1) outperforms Diffusion Policy (NFE = 50) and Flow Matching Policy (NFE = 10 & 1), achieving higher success rates and faster speed, demonstrating superior effectiveness.

| Method | NFE | Spatial (%) | Object (%) | Goal (%) | Long (%) | Average (%) | Speed (Hz) |
|---|---|---|---|---|---|---|---|
| OpenVLA-DP [32] | 50 | 92.0 | 75.0 | 93.4 | 11.8 | 68.1 | 0.32 |
| OpenVLA-FlowMatching∗ | 1 | 95.0 | 97.6 | 96.0 | 85.2 | 93.5 | 5.92 |
| OpenVLA-FlowMatching∗ | 10 | 96.0 | 97.2 | **97.8** | 83.6 | 93.7 | 1.26 |
| OpenVLA-FreqPolicy (**Ours**) | 1 | **97.0** | **98.6** | 96.0 | **87.6** | **94.8** | **6.05** |

OpenVLA-DP and OpenVLA-FlowMatching in all categories except for the Goal suite, where it is tied for second-best. Compared to OpenVLA-DP, which has an average success rate of 68.1%, OpenVLA-FreqPolicy shows a significant improvement, especially in the Long task suite (87.6% vs. 11.8%). Notably, OpenVLA-FreqPolicy with just 1 NFE achieves a higher average success rate than OpenVLA-FlowMatching with 10 NFE (94.8% vs. 93.7%), while offering significantly faster inference speed (5 times faster).

## 4.2 Real World Experiments

**Task Design.** We design 3 long-horizon real-world tasks on 2 different robotic arms as illustrated in Fig. 3, to evaluate the FreqPolicy: 1) **Fruit Sorting Task**. The Franka robot is required to sequentially place a banana, an avocado, and a mango into a basket. This is a compound pick-and-place task involving ordered object manipulation. 2) **Toy Organization Task.** The Franka robot must first pull open a drawer, place two toy dolls inside, and then close the cabinet door. This task integrates multiple skills, including pull, push, and pick-and-place. 3) **Trash Disposal Task.** The UR robot needs to open a trash bin lid, place several pieces of food waste (e.g., half a bun and half a piece of bread) into the bin, and then close the lid. This task requires a combination of pressing and pick-and-place fine manipulation capabilities. More demonstrations can be found in Sec. A.5.

**Main Results.** We primarily compare against Diffusion Policy [11] and Flow Matching Policy [59, 12, 19] with 1-step and 10-step inference settings, respectively. For each evaluation, we perform 20 trials with various initializations on the physical robot and report the mean success rate. We also measure and report the real-time inference frequency of each policy using an NVIDIA RTX 4090. As shown in Fig. 3a and Fig. 3b, FreqPolicy achieves success rates comparable to, and in some cases exceeding, those of multi-step policy models, while using only single-step inference. For example, on Task 1, FreqPolicy achieves a top success rate of 70% at an inference speed of 93.5 Hz, outperforming multi-step Diffusion Policy (55% at 19.8 Hz) and Flow Matching Policy (60% at 20.2 Hz). Similar trends are observed across all tasks. These results demonstrate that FreqPolicy can match or surpass the performance of multi-step policy models with just one inference step, while significantly reducing computational overhead. Additional evaluation details are provided in Sec. A.5.

## 4.3 Ablation Study

Table 4 shows the effect of various consistency constraints on one-step action prediction. Model #1 serves as a baseline, using vanilla flow matching for one-step action generation. Model #2 equips

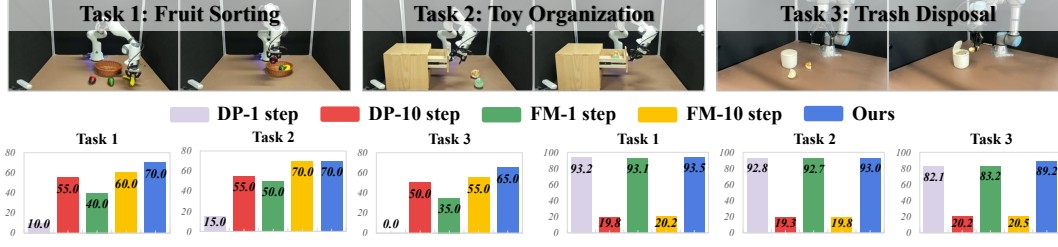

(a) Success rate (%) on three real-world tasks.    (b) Inference speed (Hz) on three real-world tasks.

Figure 3: **Demonstrations of three real-world tasks (*top*) and the test results of different policies (*bottom*).** We evaluated the success rate (%) and inference speed (Hz) for all methods. DP stands for Diffusion Policy, FM for Flow Matching Policy, and the number represents inference steps of the corresponding policy model. FreqPolicy consistently outperformed baselines in both success rate and inference speed, demonstrating its effectiveness on real-world robotic platforms.

Table 4: **Ablations on Robomimic**. We analyze the impact of various designs in transitioning from a vanilla flow matching policy with one-step action generation to our FreqPolicy. Results on the Lift are omitted, as all methods achieve saturated performance. In addition, we evaluate FreqPolicy under different spectral supervision settings to assess the importance of frequency-aware consistency.

| Model | Consistency Objectives | Can | Square | Transport | Toolhang |
|-------|------------------------|-----|--------|-----------|----------|
| # 1 | Vanilla Flow Matching (NFE=1) | 0.94 | 0.90 | 0.84 | 0.76 |
| # 2 | *w/* consistency constrain | **0.98** | 0.92 | 0.90 | 0.78 |
| # 3 | *w/* frequency consistency constrain - high | 0.96 | 0.92 | 0.88 | 0.80 |
| # 4 | *w/* frequency consistency constrain - low | 0.95 | 0.90 | 0.88 | 0.82 |
| # 5 | *w/* frequency consistency constrain - full | 0.96 | 0.92 | 0.92 | 0.82 |
| # 6 | *w/* frequency consistency constrain - adaptive | 0.97 | **0.93** | **0.92** | **0.88** |

Model #1 with a consistency loss similar to Consistency-FM [68], yielding a modest improvement (e.g., success rate increases to 90% on the transport task).

To investigate the effect of supervising different frequency bands, Model #3 and Model #4 introduce low-frequency and high-frequency consistency constraints on top of Model #1, respectively. These methods manually select specific frequency components for constraints and result in minor improvements. We attribute this to the diverse and dynamic nature of frequency components in action chunks during robotic manipulation. Model #5 applies consistency constraints across the full frequency component, leading to a more substantial gain, raising transport from 90% to 92% and toolhang from 78% to 82%, thereby validating the efficacy of frequency consistency. Finally, Model #6 incorporates our adaptive frequency loss, further boosting performance to state-of-the-art levels and confirming the advantage of adaptive frequency constraints.

## 5   Limitation & Conclusion

In this work, we introduce FreqPolicy, a novel one-step action generation policy that performs robustly under both 2D and 3D inputs. For the first time, FreqPolicy imposes a frequency-domain consistency constraint on flow matching, encouraging consistent velocities across different timesteps, and incorporates an adaptive loss that emphasizes higher-variance frequency components. Through this training regimen, FreqPolicy learns a straight-line flow to produce single-step actions. We validate its effectiveness on 53 simulated tasks and demonstrate a significant speed-up on 40 Libero tasks for a VLA model. We hope that FreqPolicy will advance embodied vision–motion planning and VLA models in real-world applications. However, our work presents certain limitations. We primarily validated the effectiveness of our proposed FreqPolicy on flow matching. In principle, the same core idea can be extended to diffusion-based visual-motor policies, which we will systematically investigate in future work. Our method demands more computational resources, as it requires twice forward passes to process two random samples. In future research, we will explore the computationally efficient one-step action generators.

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

# A Technical Appendices and Supplementary Material

In this appendix, we first present further details on the model architecture of FreqPolicy in Sec. A.1. Next, Sec. A.2 provides additional visualizations and frequency-domain analyses of various action chunks. We then supplement the simulation experiments with additional information in Sec. A.3, along with details on integrating FreqPolicy into VLA models in Sec. A.4. Finally, Sec. A.5 describes the setup and results of the real-world experiments.

## A.1 More Model Configuration

**As a Visuomotor Policy.** Following previous works [11, 76, 28, 42, 78], we adopt a standard 1D CNN-based U-Net architecture as the backbone of FreqPolicy to ensure a fair comparison with existing models. FreqPolicy is designed to accept both 2D and 3D observations as input. For 2D images, we employ ResNet-18 as the visual encoder. For 3D input, we follow [76, 78] and use a lightweight MLP to encode the input point cloud.

**As a Head of the VLA Model.** FreqPolicy can also serve as a policy head for existing VLA models, as long as the underlying VLA is capable of predicting action vector fields. In this work, we integrate FreqPolicy into OpenVLA [33], following the setup in [32], where the predicted action vector field is obtained by applying a nonlinear mapping to the noise-conditioned latent features.

## A.2 Extended Frequency Analysis for Robotic Manipulation

In Fig. 4, we present visualizations of action chunks from different real-world and simulation scenarios, including the observed images, the corresponding multi-dimensional temporal action signals, and the transformed DCT frequency coefficients. In the simulation scenario shown in Fig. 4a, the Franka robot transitions from "approaching the blue can" to "grasping the can", during which the expected action chunk contains high-frequency variations associated with gripper motion. Similarly, in Fig. 4b, as the robot moves from "approaching the stove" to "placing down the kettle", the gripper opening action introduces high-frequency signals, while other control signals remain relatively smooth. In the real-world scenario shown in Fig. 4c, during the process of picking up the doll, the Franka robot exhibits relatively smooth motion transitions, with the action signal primarily dominated by low-frequency components. Hence, based on the above observations of different action chunks and their frequency-domain characteristics, we justify the motivation of the adaptive frequency coefficient loss introduced in Sec 3. This loss enables the model to focus more effectively on the frequency components that exhibit meaningful variation across diverse action chunks.

## A.3 More Details on Visuomotor Policy Simulation

### A.3.1 More Details on Simulation with 2D Inputs

**Implementation Details.** We implement the RectifiedFlow [40, 38], ConsistencyPolicy [53], and Consistency-FM [68] to comprehensively evaluate our proposed FreqPolicy. For RectifiedFlow, we adopt a 1D CNN-based U-Net to predict action vector fields from observations. Following [6], we sample time steps $t$ from a Beta distribution during training to interpolate intermediate states. For ConsistencyPolicy, we follow the procedure in [53], where an EDM [29] teacher model is first trained in the initial stage, and then a student model is distilled for one-step action generation using the CTM objective [31]. For Consistency-FM, we also use a 1D CNN-based U-Net to predict action vector fields from observations. We then sample different time steps $r$ and $s$ from a uniform schedule and apply velocity field consistency constraints between them. To train all models, we use observations from the past 2 time steps as input and predict a 16-step action chunk, from which the first 8 steps are selected for execution. All models are trained using a batch size of 128 with the AdamW optimizer and a learning rate of 1.0e-4. Training is conducted for 1000 epochs on a single NVIDIA A100 GPU. For ConsistencyPolicy, the student model is trained for 450 epochs.

### A.3.2 More Details on Simulation with 3D Inputs

**Implementation Details.** Following the prior works [76, 78], we use a lightweight MLP to encode the point cloud and use a 1D CNN-based U-Net to predict the action vector field. To train the FreqPolicy model, we use a batch size of 128 and the AdamW optimizer with a learning rate of

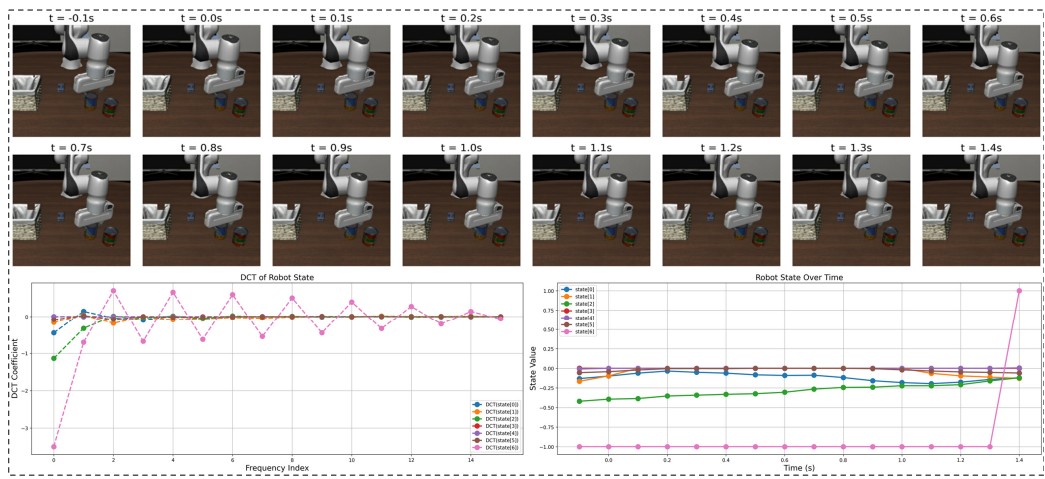

(a) Visualization of an action chunk from the Franka simulation dataset. When the robot transitions from "reach the blue can" to "grasping it", the change in gripper introduces high-frequency signal variations. Other states exhibit no significant changes and tend to be smoother.

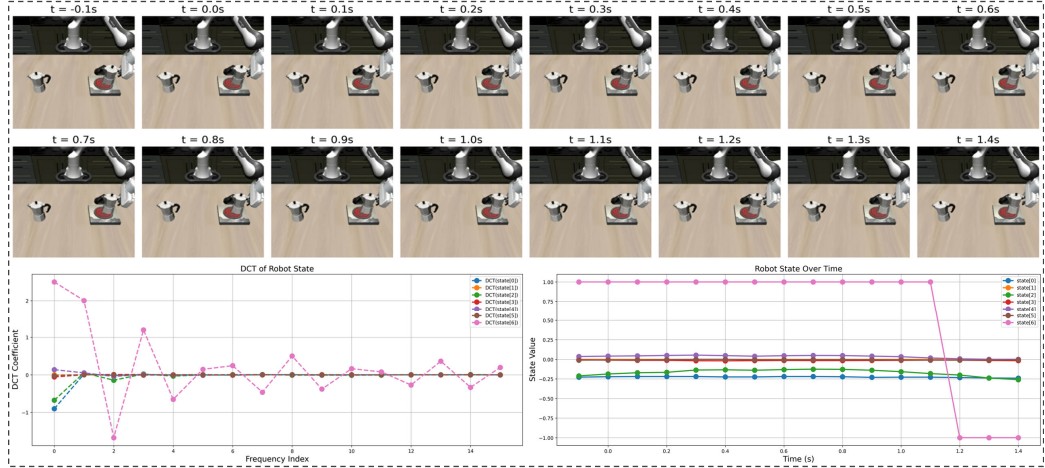

(b) Visualization of an action chunk from the Franka simulation dataset. During the transition from "approaching the stove" to "placing down the kettle", the gripper exhibits high-frequency variations, while other states remain nearly unchanged.

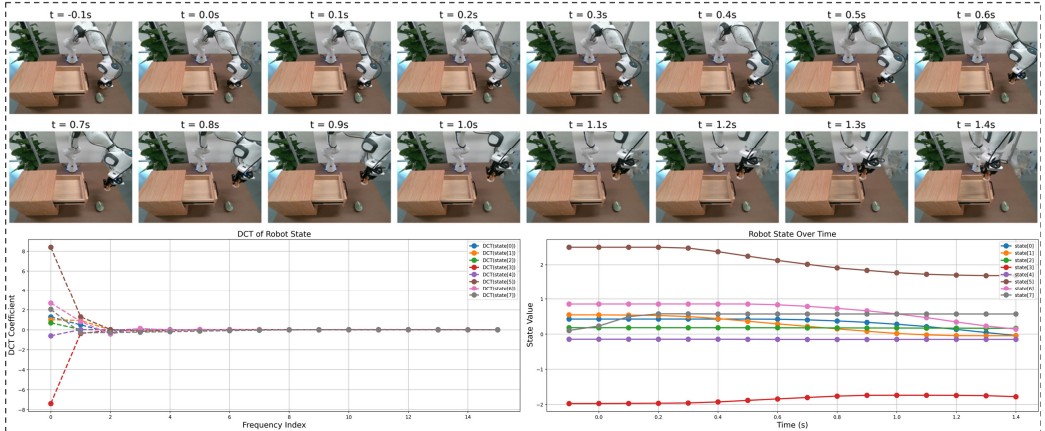

(c) Visualization of an action chunk from the Franka robot demonstrations. Once the robot successfully grasps the doll and begins placing it into the drawer, the overall motion exhibits relatively slow changes under a 30 Hz action sampling rate, dominated by low-frequency signals.

Figure 4: Visualization of the spectral and temporal signals across different action chunks.

Table 5: We report the evaluation details of the 53 challenging tasks from Adroit [55] and Meta-World [74] under 3 random seeds, and report the mean success rate (%) and standard deviation for each task. Tasks marked with an asterisk ∗ indicate re-implemented versions. Compared to ManiCM [42], a one-step action generation model using consistency distillation based on diffusion-based policies, our method achieves superior performance. Similarly, we achieve an overall improvement of 9.4% over SDM [28] that adopts variational score distillation [72, 71] to achieve a one-step diffusion policy. Finally, in comparison to the flow-based policy FlowPolicy [78], which uses only spatial-domain consistency loss, FreqPolicy still leads by 2.8% gains, demonstrating the effectiveness of our frequency consistency constraint.

| Alg / Task | Adroit | | | Meta-World (Easy) | | |
|---|---|---|---|---|---|---|
| | Adroit Hammer | Adroit Door | Adroit Pen | Button Press | Coffee Button | Plate Slide Back Side |
| Diffusion Policy | 45 ± 5 | 37 ± 2 | 13 ± 2 | 99 ± 1 | 99 ± 1 | 100 ± 0 |
| 3D Diffusion Policy | 100 ± 0 | **75** ± 3 | 48 ± 3 | 100 ± 0 | 100 ± 0 | 100 ± 0 |
| ManiCM | 100 ± 0 | 68 ± 1 | 49 ± 4 | 100 ± 0 | 100 ± 0 | 100 ± 0 |
| SDM Policy | **100** ± 0 | 73 ± 2 | 49 ± 4 | 100 ± 0 | 100 ± 0 | 100 ± 0 |
| FlowPolicy∗ | 97 ± 3 | 62 ± 6 | 49 ± 5 | 100 ± 0 | 100 ± 0 | 100 ± 0 |
| Ours | 98 ± 2 | 68 ± 5 | **52** ± 4 | **100** ± 0 | **100** ± 0 | **100** ± 0 |

| Alg / Task | Meta-World (Easy) | | | | | |
|---|---|---|---|---|---|---|
| | Button Press Topdown | Button Press Topdown Wall | Button Press Wall | Peg Unplug Side | Door Close | Door Lock |
| Diffusion Policy | 98 ± 1 | 96 ± 3 | 97 ± 3 | 74 ± 3 | 100 ± 0 | 86 ± 8 |
| 3D Diffusion Policy | 99 ± 1 | 96 ± 3 | 100 ± 0 | **93** ± 3 | 100 ± 0 | 96 ± 3 |
| ManiCM | 100 ± 0 | 96 ± 2 | 98 ± 3 | 71 ± 15 | 100 ± 0 | 98 ± 2 |
| SDM Policy | 98 ± 2 | 99 ± 1 | 100 ± 0 | 74 ± 19 | 100 ± 0 | 96 ± 2 |
| FlowPolicy∗ | 100 ± 0 | 100 ± 0 | 100 ± 0 | 88 ± 5 | 100 ± 0 | 100 ± 0 |
| Ours | **100** ± 0 | **100** ± 0 | **100** ± 0 | 87 ± 4 | **100** ± 0 | **100** ± 0 |

| Alg Task | Meta-World (Easy) | | | | | | | |
|---|---|---|---|---|---|---|---|---|
| | Door Open | Door Unlock | Drawer Close | Drawer Open | Faucet Close | Faucet Open | Handle Press | Handle Pull |
| Diffusion Policy | 98 ± 3 | 98 ± 3 | 100 ± 0 | 93 ± 3 | 100 ± 0 | 100 ± 0 | 81 ± 4 | 27 ± 22 |
| 3D Diffusion Policy | 100 ± 0 | 100 ± 0 | 100 ± 0 | 100 ± 0 | 100 ± 0 | 100 ± 0 | 100 ± 0 | **52** ± 8 |
| ManiCM | 100 ± 0 | 82 ± 16 | 100 ± 0 | 100 ± 0 | 100 ± 0 | 100 ± 0 | 100 ± 0 | 10 ± 10 |
| SDM Policy | 100 ± 0 | 100 ± 0 | 100 ± 0 | 100 ± 0 | 99 ± 1 | 100 ± 0 | 100 ± 0 | 28 ± 11 |
| FlowPolicy∗ | 100 ± 0 | 100 ± 0 | 100 ± 0 | 100 ± 0 | 99 ± 0 | 100 ± 0 | 100 ± 0 | 25 ± 8 |
| Ours | **100** ± 0 | **100** ± 0 | **100** ± 0 | **100** ± 0 | **100** ± 0 | **100** ± 0 | **100** ± 0 | 22 ± 5 |

| Alg Task | Meta-World (Easy) | | | | | | | |
|---|---|---|---|---|---|---|---|---|
| | Handle Press Side | Handle Pull Side | Lever Pull | Plate Slide | Plate Slide Back | Dial Turn | Reach | Reach Wall |
| Diffusion Policy | 100 ± 0 | 23 ± 17 | 49 ± 5 | 83 ± 4 | 99 ± 0 | 63 ± 10 | 18 ± 2 | 59 ± 7 |
| 3D Diffusion Policy | 0 ± 0 | **82** ± 5 | **84** ± 8 | **100** ± 0 | 100 ± 0 | 91 ± 0 | 26 ± 3 | 74 ± 3 |
| ManiCM | 0 ± 0 | 48 ± 11 | 82 ± 7 | 100 ± 0 | 96 ± 5 | 84 ± 2 | 33 ± 3 | 62 ± 5 |
| SDM Policy | 0 ± 0 | 68 ± 6 | 84 ± 9 | 100 ± 0 | 100 ± 0 | 88 ± 3 | **34** ± 3 | **80** ± 1 |
| FlowPolicy∗ | 100 ± 0 | 50 ± 6 | 74 ± 5 | 95 ± 3 | 100 ± 0 | 81 ± 4 | 29 ± 10 | 69 ± 6 |
| Ours | **100** ± 0 | 59 ± 6 | 80 ± 5 | 96 ± 2 | **100** ± 0 | **91** ± 5 | 32 ± 10 | 72 ± 5 |

| Alg Task | Meta-World (Easy) | | | Meta-World (Medium) | | | | |
|---|---|---|---|---|---|---|---|---|
| | Plate Slide Side | Window Close | Window Open | Basketball | Bin Picking | Box Close | Coffee Pull | Coffee Push |
| Diffusion Policy | 100 ± 0 | 100 ± 0 | 100 ± 0 | 85 ± 6 | 15 ± 4 | 30 ± 5 | 34 ± 7 | 67 ± 4 |
| 3D Diffusion Policy | 100 ± 0 | 100 ± 0 | 99 ± 1 | **100** ± 0 | **56** ± 14 | 59 ± 5 | 79 ± 2 | 96 ± 2 |
| ManiCM | 100 ± 0 | 100 ± 0 | 80 ± 26 | 4 ± 4 | 49 ± 17 | **73** ± 2 | 68 ± 18 | 96 ± 3 |
| SDM Policy | 100 ± 0 | 100 ± 0 | 78 ± 18 | 28 ± 26 | 55 ± 13 | 61 ± 3 | 72 ± 9 | **97** ± 2 |
| FlowPolicy∗ | 100 ± 0 | 100 ± 0 | 100 ± 0 | 85 ± 7 | 45 ± 7 | 56 ± 4 | **89** ± 3 | 94 ± 2 |
| Ours | **100** ± 0 | **100** ± 0 | **100** ± 0 | 77 ± 4 | 40 ± 7 | 61 ± 5 | 84 ± 5 | 95 ± 3 |

| Alg Task | Meta-World (Medium) | | | | | | Meta-World (Hard) | | |
|---|---|---|---|---|---|---|---|---|---|
| | Hammer | Peg Insert Side | Push Wall | Soccer | Sweep | Sweep Into | Assembly | Hand Insert | Pick Out of Hole |
| Diffusion Policy | 15 ± 6 | 34 ± 7 | 20 ± 3 | 14 ± 4 | 18 ± 8 | 10 ± 4 | 15 ± 1 | 0 ± 0 | 0 ± 0 |
| 3D Diffusion Policy | **100** ± 0 | 79 ± 4 | 78 ± 5 | 23 ± 4 | **92** ± 4 | **38** ± 9 | 100 ± 0 | **28** ± 8 | **44** ± 3 |
| ManiCM | 98 ± 2 | 75 ± 8 | 31 ± 7 | 27 ± 3 | 54 ± 16 | 37 ± 13 | 87 ± 3 | 28 ± 15 | 30 ± 16 |
| SDM Policy | 98 ± 2 | **83** ± 5 | **83** ± 4 | 25 ± 2 | 90 ± 6 | 32 ± 15 | 100 ± 0 | 24 ± 14 | 34 ± 24 |
| FlowPolicy∗ | 97 ± 3 | 69 ± 5 | 56 ± 6 | 24 ± 6 | 91 ± 3 | 26 ± 5 | 91 ± 2 | 23 ± 5 | 33 ± 3 |
| Ours | 95 ± 2 | 70 ± 4 | 64 ± 11 | **27** ± 4 | 88 ± 4 | 23 ± 4 | **100** ± 0 | 18 ± 3 | 40 ± 4 |

| Alg Task | Meta-World (Hard) | | | Meta-World (Very Hard) | | | | | |
|---|---|---|---|---|---|---|---|---|---|
| | Pick Place | Push | Push Back | Shelf Place | Disassemble | Stick Pull | Stick Push | Pick Place Wall | **Average** |
| Diffusion Policy | 0 ± 0 | 30 ± 3 | 0 ± 0 | 11 ± 3 | 43 ± 7 | 11 ± 2 | 63 ± 3 | 5 ± 1 | 55.5 ± 3.58 |
| 3D Diffusion Policy | 0 ± 0 | 56 ± 5 | 0 ± 0 | 47 ± 2 | **91** ± 4 | 67 ± 0 | 100 ± 0 | 74 ± 4 | 76.1 ± 2.32 |
| ManiCM | 0 ± 0 | 55 ± 2 | 0 ± 0 | 48 ± 3 | 87 ± 3 | 63 ± 2 | 100 ± 0 | 37 ± 16 | 69.0 ± 4.60 |
| SDM Policy | 0 ± 0 | 57 ± 0 | **100** ± 0 | 51 ± 4 | 86 ± 10 | 68 ± 10 | 0 ± 0 | 53 ± 12 | 74.8 ± 4.51 |
| FlowPolicy∗ | 57 ± 4 | 59 ± 5 | - | 47 ± 6 | 74 ± 5 | 66 ± 7 | 100 ± 0 | **92** ± 4 | 77.2 ± 2.84 |
| Ours | **63** ± 5 | **60** ± 5 | - | **51** ± 3 | 82 ± 6 | **74** ± 6 | **100** ± 0 | 90 ± 4 | **78.5 ± 2.61** |

1.0e-4, training for 3000 epochs. Evaluation is conducted every 200 epochs, and the best-performing checkpoint is saved. All experiments are conducted on a single NVIDIA A100 GPU.

**Result Details.** We report the detailed success rates of FreqPolicy on 53 tasks from the Adroit and MetaWorld benchmark in Table 5. The success rate of each task is averaged over experiments conducted with 3 random seeds.

## A.4 More Details about VLA Settings

**Implementation.** For Diffusion Policy, we leverage the implementation of 'OpenVLA (fine-tuned) + PD&AC, Cont-Diffusion' from OpenVLA-OFT (OpenVLA-DP in this paper). For the Flow Matching Policy, we integrate it into the OpenVLA pipeline (i.e., OpenVLA-FlowMatching) based on the original implementation [40]. We train for 150K gradient steps for all models using a batch size of 16 across 4 A100 GPUs, policies receive one third-person image, one wrist camera image, robot proprioceptive state, and language instruction as input. The detailed hyperparameters are shown in Table 6. Please note that our training settings are not consistent with the original paper. In the default settings of OpenVLA-OFT [32], they train for 150K steps, using a batch size of 64 across 8 GPUs. However, in our experiments, considering the computational overhead and resource capabilities, we did not follow the full training settings consistent with the original paper (for example, the results we show are all trained on 2.4M samples, while the original paper trained 9.6M samples or even more), but to show the compatibility and superiority with VLA in comparison with the multi-step policy baselines. In the test, we perform 50-steps inference for OpenVLA-DP, and 1-step and 10-steps inference for OpenVLA-FlowMatching. The inference speed is averaged over the first three episodes of LIBERO-spatial to estimate performance. We follow the default settings in OpenVLA-OFT [4] for other configurations.

Table 6: **The hyperparameter settings for VLA experiments.** These values are kept consistent across all methods.

| Parameter | Values |
|---|---|
| use_film | True |
| use_proprio | True |
| num_images_in_input | 2 |
| lora_rank | 32 |
| image_aug | True |
| num_steps_before_decay | 100,000 |
| max_steps | 150,000 |
| num_gpus | 4 |
| batch_size | 4 |
| learning_rate | 5e-4 |
| num_trials_per_task | 50 |
| num_inference_steps | 1 or 10 or 50 |

**Compare with OpenVLA-OFT.** OpenVLA-OFT is an improved version of OpenVLA [33] that integrates parallel decoding, action chunking, continuous action representation, and an L1 regression-based learning objective. It achieves state-of-the-art performance on the LIBERO simulation benchmark, significantly improving the average success rate while increasing action generation throughput by 26 times. We reproduced OpenVLA-OFT [32] under the same training and testing settings, and the results are shown in Table 7. Specifically, our method leads or is on par with OpenVLA-OFT in all categories except LIBERO-10, and leads OpenVLA-OFT in average success (94.8% vs. 94.2%). Moreover, even though FreqPolicy leverages an additional policy head, the inference speed remains competitive. (6.05Hz vs. 6.15Hz).

---

[4] https://github.com/moojink/openvla-oft

Table 7: **Comparison with OpenVLA-OFT on LIBERO simulation benchmark.**

| Method | NFE | Spatial (%) | Object (%) | Goal (%) | Long (%) | Average (%) | Speed (Hz) |
|---|---|---|---|---|---|---|---|
| OpenVLA-DP | 50 | 92.0 | 75.0 | 93.4 | 11.8 | 68.1 | 0.32 |
| OpenVLA-FlowMatching* | 1 | 95.0 | 97.6 | 96.0 | 85.2 | 93.5 | 5.92 |
| OpenVLA-FlowMatching* | 10 | 96.0 | 97.2 | 97.8 | 83.6 | 93.7 | 1.26 |
| OpenVLA-OFT | - | 96.6 | 98.6 | 91.0 | 90.6 | 94.2 | 6.15 |
| OpenVLA-FreqPolicy | 1 | 97.0 | 98.6 | 96.0 | 87.6 | 94.8 | 6.05 |

## A.5 More Details about Real-World Tasks

**Implementation.** The real-world experiments are based on the open-source LeRobot framework [5]. LeRobot is designed to support real-world robotics research by providing models, datasets, and tools in PyTorch. It includes state-of-the-art methods, such as Diffusion Policy, which we use as a baseline, that have demonstrated strong transfer capabilities to real-world settings, with a focus on imitation learning. We convert our collected real-world data into the LeRobot-supported format and integrate both the Flow Matching Policy and our proposed FreqPolicy into the framework. This enables training directly on real data, facilitating practical deployment and inference. Training was conducted on a single NVIDIA A100 GPU. To ensure a fair comparison, all policy models were trained with aligned hyperparameters provided in Table 8. We also provide additional demonstration images for the three tasks in Fig. 5, as well as demonstration videos in the supplementary materials to facilitate better understanding.

Table 8: **The hyperparameter settings for real-world experiments.** These values are kept consistent across all methods.

| Parameter | Values |
|---|---|
| input_images_Franka | [camera_front, camera_wrist] |
| input_images_UR | [camera_left, camera_wrist] |
| use_robot_state | True |
| image_size | $480 \times 480$ |
| vision_backbone | ResNet-18 |
| n_obs_steps | 2 |
| horizon | 48 |
| n_action_steps | 48 |
| num_inference_steps | 1 or 10 |
| batch_size | 64 |
| # of training steps | 100,000 |
| optimizer | Adam |
| learning_rate | 1e-4 |
| weight_decay | 1e-6 |
| grad_clip_norm | 10.0 |

**Evaluation Criteria.** The initialization of the task environment plays a critical role, as it directly affects whether the robot can complete a task under the guidance of the policy model. Moreover, the conditions leading to task failure vary across different tasks. To facilitate a deeper understanding of the real-world experiments presented in this work, we provide target and object placement rules for the three tasks, along with detailed descriptions of the conditions under which task execution fails:

**1) Fruit Sorting.**
   *Target:* Pick up bananas, avocados, and mangoes and put them into the basket in order.
   *Object placement rules:* During execution, the fruit basket remains stationary in a fixed position. The spatial order of the three fruits—banana, avocado, and mango, from right to left—remains unchanged. However, in each trial, the exact placement of the fruits is randomly initialized within the front half of the basket.

---

[5] https://github.com/huggingface/lerobot

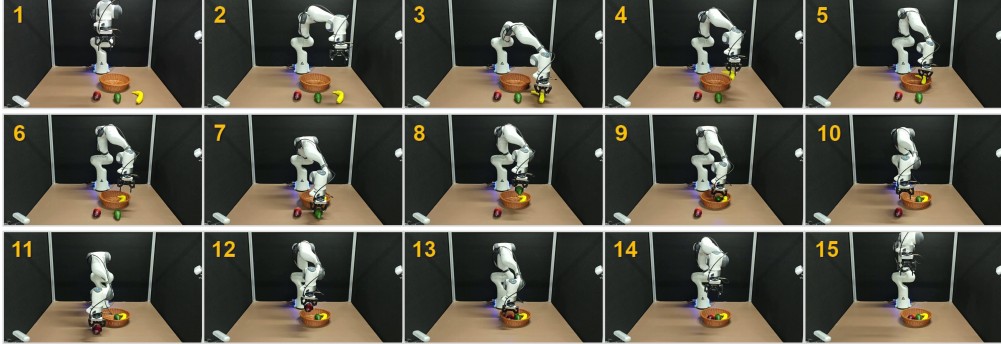

(a) Demonstrations of real-world **Task_1**. This task is carried out on the Franka platform. The robot arm is tasked with sequentially picking up three different fruits—banana, avocado, and mango—and placing them into a basket. The fruits are positioned one at a time in random locations in front of the basket, and the task is deemed successful only if all three fruits are correctly picked up and placed in the basket.

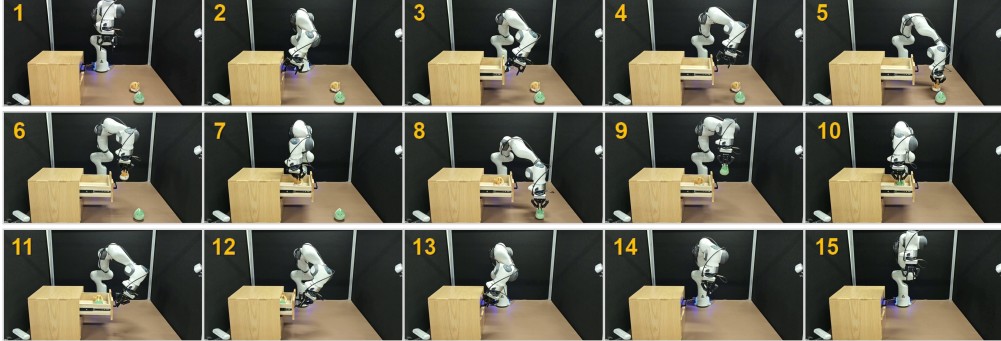

(b) Demonstrations of real-world **Task_2**. This task is carried out on the Franka platform. The robot arm must first rotate to the appropriate angle, open its gripper, and insert it into the drawer handle to pull the drawer open. It then sequentially picks up two plush toys placed randomly on the table and places them inside the drawer. Finally, it closes the gripper and pushes the handle to shut the drawer. This requires the robot to accurately perceive the drawer handle's start and end positions and be robust to the random positions of the toys.

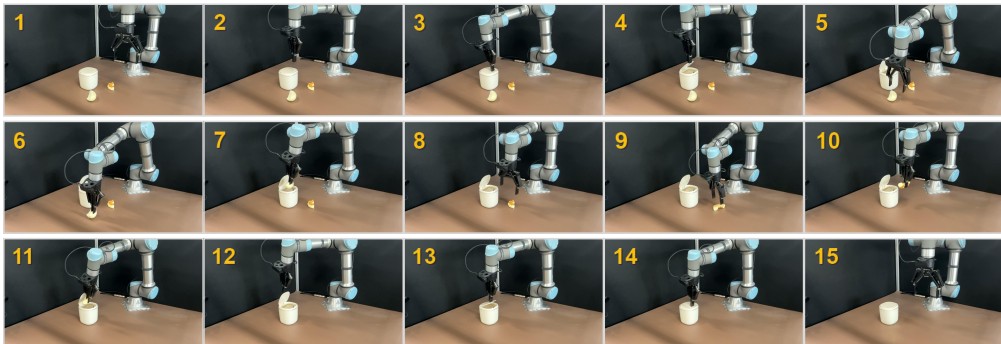

(c) Demonstrations of real-world **Task_3**. This task is carried out on the UR platform. The robot arm first closes its gripper and attempts to tap a specific spot on the trash bin lid with the tip of the gripper to trigger it to pop open. Then, with the gripper open, the arm sequentially picks up two pieces of soft food waste—a half-steamed bun and a piece of bread—randomly placed on the table. Finally, the gripper closes again, moves behind the lid, and pushes it while tapping once more to fully close the bin. This process requires precise tapping actions and robustness to the random placement of the food waste.

Figure 5: **Demonstrations of three long-horizon real-world tasks.** Following a consistent protocol, 300 episodes are collected for each task for training, after which the trained policies are evaluated. The numbers in each image indicate the sequential steps in the task execution process.

*Cases of task execution failed:*
- If any fruit falls while picking it up, it will be considered a failure case.
- If the same fruit fails to be picked up more than 3 times in total, it will be considered a failure case.
- If all fruits are not completely put into the basket, it will be considered a failure case.

**2) Toy Organization.**
*Target:* Open the drawer, then put the two dolls into the drawer one by one, and close the drawer.
*Object placement rules:* During each execution, the cabinet remains in a fixed position, while two dolls are reinitialized with their relative spatial positions preserved.
*Cases of task execution failed:*
- If the drawer is not successfully opened, it is considered a failure case.
- If the drawer is not closed after all the dolls are placed, it is considered a failure case.
- If all the dolls are not placed in the drawer, it is considered a failure case.
- If the cumulative number of failures to pick up the same doll is 3 or more, it is considered a failure case.

**3) Trash Disposal.**
*Target:* Tap the trash can lid to make it pop open, put the food waste (steamed buns and bread) into the trash can one by one, and tap the lid again to close it.
*Object placement rules:* During the execution, the position of the trash can is fixed, the relative spatial positions of the two types of food waste remain unchanged, and the trash positions are reinitialized for each execution.
*Cases of task execution failed:*
- If the lid is not closed after picking up the food waste, it is considered a failure case.
- If the number of failed attempts to pick up the same food waste exceeds 3 times, the machine will be deemed as a failed case.
- If the lid is closed before all the food wastes are put into the trash can, it will be considered a failure case.
- If the lid is not opened and food waste is taken, it will be deemed as a failure case.

