# OpenReview forum: "FreqPolicy: Efficient Flow-based Visuomotor Policy via Frequency Consistency"
_NeurIPS.cc/2025/Conference — NeurIPS 2025 poster_

### Official Review · Reviewer_4VTk · 2025-07-02

**Clarity:** 4
**Significance:** 3
**Originality:** 3
**Rating:** 5
**Confidence:** 3

**Summary:**

The paper proposes a one-step visuomotor policy that incorporates temporal knowledge for robotic manipulation. The authors introduce two key innovations: (1) a Frequency Consistency Objective, which enforces consistency between the velocities of action segments across different timesteps in the frequency domain; and (2) an Adaptive Frequency Component Loss, which dynamically emphasizes frequency components with greater discrepancies through adaptive weighting. The writing is clear, and the effectiveness of the approach is validated on MetaWorld, demonstrating superiority over existing one-step visuomotor policies and achieving significant improvements in inference speed for VLA models.

**Questions:**

Methodology

- Equation 11 employs the type-II DCT. Have the authors conducted ablation studies using alternative similarity functions?
- Equation 1 enforces the velocity field as a straight trajectory. How sensitive is the proposed objective function to the hyperparameters, like the ratio with respect to L_fm ?

**Ethical Concerns:**

["NO or VERY MINOR ethics concerns only"]

**Final Justification:**

Thanks for your detailed response. I have no further questions.

**Limitations:**

yes

**Quality:**

3

**Strengths And Weaknesses:**

Quality

The paper presents a well-motivated idea, supported by comprehensive experimental results. The writing is detailed, and the visualizations are clear. The experiments are thorough and effectively demonstrate the validity of the proposed innovations.

Clarity

The paper is clearly written and easy to follow. The figures are well-presented, and the experimental results and ablation studies are clearly designed to validate the proposed innovations.


Significance

The experimental results demonstrate the effectiveness of the proposed approach. The flow-matching-based policy model is of interest to both academia and industry, and the proposed objective function is broadly applicable to sensorimotor policy networks and VLA models.

It is strong due to its comprehensive experiments including real-world experiments and reasonable innovation design.

Originality

The paper demonstrates strong originality, and the core idea is precise and impactful.

---

> ### Author Rebuttal · Authors · 2025-07-31
>
> We sincerely thank you for the thorough review and comments on our work. Please find our responses below:
>
>
> ------
> **Q1:** *Have the authors conducted ablation studies using alternative similarity functions like DCT-II?*
>
> **A:** Thanks for your valuable question. In the version of FreqPolicy we reported, we initially applied the classic DCT‑II transform to project the time‑series action signals into the frequency domain, which is a common effective approach. Per your suggestion, we further investigated the impact of different frequency transforms on FreqPolicy’s performance by selecting two additional methods: the FFT transform and the DCT‑I transform.
>
> Formally, using a DCT-I transform differs little from DCT-II: one needs only swap the transform operator. With an FFT, the output is a complex spectrum rather than the real coefficients of a DCT. However, since FreqPolicy cares only about the distribution of frequency components and not signal reconstruction, we compute the consistency constraint using only the real parts of the FFT coefficients.
>
> To balance time constraints with experimental rigor, we conducted ablation studies on a subset of MetaWorld tasks. Specifically, we evaluated all 11 tasks from the most challenging Hard and Very Hard splits. For each task, we ran experiments with 3 random seeds (consistent with our main paper's evaluation protocol). We validated every 200 training epochs and averaged the top‑5 checkpoint success rates per seed. Finally, we report the overall mean success rate and standard deviation across all tasks for each method in the table below.
>
> |                 | Vanilla       | FFT         | DCT‑I        | DCT‑II (Ours)        |
> | :-------------: | ------------- | ----------- | ------------ | ------------- |
> | Avg. Results | 0.5935±0.0422 | 0.619±0.035 | 0.6142±0.039 | 0.6148±0.0382 |
>
> From these results, we observe that FreqPolicy is not sensitive to the choice of frequency-domain transform: whether using the classic FFT or various DCT variants, the final performance has changed very little. This is unsurprising given that these transforms are essentially very similar.
>
> Meanwhile, the results of FFT, DCT-I, and DCT-II are better than the vanilla baseline. This result indicates that regardless of the method used to process the time-domain signals, there is a consistent improvement in manipulation tasks. It supports our motivation that leveraging robotic learning with time-domain information is effective.
>
> We will include these additional experiments and explanations in the revision.
>
> ------
> **Q2:** *How sensitive is the proposed objective function to hyperparameters, like the ratio with respect to $\mathcal{L}_{\text{fm}}$ ?*
>
> **A:** Following your suggestion, we performed a hyperparameter sensitivity analysis on the loss defined in Equation 15.  Originally, our total loss is $\mathcal{L}\_{\text{total}} = \mathcal{L}\_{\text{freq}} + \mathcal{L}\_{\text{fm}}$. We introduced a hyperparameter $\lambda$ for $\mathcal{L}\_{\text{fm}}$, yielding the final objective $\mathcal{L}\_{\text{total}} = \mathcal{L}\_{\text{freq}} + \lambda\ \mathcal{L}\_{\text{fm}}$.
>
> Specifically, we tested a range of $\lambda$ values, 0.03, 0.1, 0.3, 0.5, 1.0, 1.2, and 10, using the same experimental setup and evaluation protocol described above. The results are shown below:
>
> |                 | $\lambda=0.03$ | $\lambda=0.1$ | $\lambda=0.3$ | $\lambda=0.6$ | $\lambda=1$   | $\lambda=1.2$ | $\lambda=10$ |
> | :-------------: | :------------- | ------------- | ------------- | ------------- | ------------- | ------------- | ------------ |
> | Avg. Results | 0.5915±0.0348  | 0.6013±0.0326 | 0.6110±0.0357 | 0.617±0.0301  | 0.6148±0.0382 | 0.6091±0.0381 | 0.6148±0.044 |
>
> As shown in the table, only when $\lambda = 0.03$ did the performance slightly degrade. The results reveal that across a range ($\lambda\in[0.1, 10]$), the success rates vary minimally, demonstrating that FreqPolicy is largely insensitive to $\lambda$ and does not require meticulous hyperparameter tuning.

---

> ### Author Response · Authors · 2025-08-08
> **Further Response**
>
> Dear Reviewer,
>
> We would like to express our sincere gratitude for your thoughtful and comprehensive review of our paper, as well as your kind and encouraging evaluation.
>
> In our rebuttal, we have carefully addressed the two concerns you raised, including conducting additional experiments and providing detailed explanations. We genuinely hope these responses have clarified your questions. We will incorporate the corresponding modifications and discussions into the revised version of the manuscript to further strengthen our work.
>
> If there are any remaining questions or further suggestions you may have, please do not hesitate to let us know — we would be truly grateful for the opportunity to continue the discussion and provide additional clarification.
>
> Once again, thank you very much for your valuable comments and for highlighting important aspects of our work. Your feedback has been instrumental in helping us improve the quality and rigor of our paper.

---

### Official Review · Reviewer_fS2J · 2025-07-03

**Clarity:** 3
**Significance:** 3
**Originality:** 2
**Rating:** 4
**Confidence:** 4

**Summary:**

This paper presents FreqPolicy, a consistency flow matching method for efficient generation of visuomotor control actions. FreqPolicy considers the setup where we train a rectified flow matching policy on some demonstration data, and would like to run efficient one-step inference with good task performance. Some background works such as Consistency Flow Matching considers velocity consistency loss given the same endpoint, and FAST considers action representation in the frequency domain with DCT. For efficient inference, FreqPolicy proposes training a flow matching policy, with consistency loss in the frequency domain with focal loss-like adaptive weighting. Experiments in various sim and real environments show that this approach gives us efficient one-step flow matching policy inference with good performance. Ablation studies validate the necessity of each design component in FreqPolicy.

**Questions:**

- Would this be equivalent to taking a DCT on the action chunks, then taking a regular consistency loss during training?

**Ethical Concerns:**

["NO or VERY MINOR ethics concerns only"]

**Final Justification:**

Overall I remain positive about this paper. I agree with the authors that saving an inverse DCT for downstream deployment is a good reason to absorb the DCT into the loss. The additional experiments provided in the rebuttal period make sense.

**Limitations:**

- Yes, the paper has adequately addressed its limitations.

**Paper Formatting Concerns:**

N/A.

**Quality:**

3

**Strengths And Weaknesses:**

- The idea of frequency domain consistency matching for efficient flow matching inference, as well as adaptive weighting, is well-motivated.
- Various experiments in real and simulated environments show that FreqPolicy achieves good results with efficient one-step inference.
- The paper compares with a comprehensive set of baselines such as iterative generation with DDPM, DDIM, flow matching policies, as well as consistency models and IMLE.
- Ablation studies show that the core ideas of frequency domain consistency loss, as well as adaptive weighting, improves performance over vanilla velocity consistency or no weighting.
- The paper combines two existing ideas (DCT for action representation, and consistency flow matching).

---

> ### Author Rebuttal · Authors · 2025-07-31
>
> We appreciate your insightful review and feedback on our paper. Below is our response:
>
> ------
> **Q1:** *Would this be equivalent to taking a DCT on the action chunks, then taking a regular consistency loss during training?*
>
> **A**: Thank you for raising this insightful question. To explore the differences between these paradigms, we conducted a series of comparative experiments with the following settings:
>
> (A) Directly using a regular consistency loss on the time-domain signals (without any DCT transform).
>
> (B) Applying a DCT on the action chunks followed by a regular consistency loss (per your suggestion). The consistency loss comprises two components: 1) enforcing that velocities at different timesteps match, and 2) enforcing that transitions from any two timesteps to a third remain identical.
>
> (C) FreqPolicy without adaptive weighting (for a fair comparison with setting B).
>
> (D) The complete FreqPolicy (Setting C with the frequency adaptive weighting strategy).
>
> To maintain experimental rigor within the time constraints, we evaluated this setup on all 11 tasks from the Hard and Very Hard splits of MetaWorld using three independent random seeds (consistent with our main paper) and report the mean success rates across seeds below.
>
>
>
>
> |   Setting    |       A       |       B       |       C       |       D       |
> |:------------:|:-------------:|:-------------:|:-------------:|:-------------:|
> | Avg. Results | 0.5935±0.0422 | 0.6080±0.0454 | 0.6101±0.0454 | 0.6148±0.0382 |
>
>
> The results show that:
>
> (1) Settings B and C yield nearly comparable performance, showing that both directly applying DCT on the action chunks and applying DCT during consistency loss (our method) are similar, as both methods effectively leverage temporal information.
>
> (2) Both setting B and C outperform setting A, confirming that frequency-domain transforms enhance the extraction of temporal features compared to time-domain consistency alone.
>
> (3) Comparing setting B and C with setting D highlights the effectiveness of the adaptive weighting strategy in FreqPolicy.
>
> Additionally, we originally chose to use the FreqPolicy auxiliary loss because it adds no extra inference time for inverting frequency-domain actions to time-domain actions compared to directly applying DCT on the action chunks.
>
> We will incorporate this detailed comparison and discussion in the revision.

---

> > ### Comment · Reviewer_fS2J · 2025-08-04
> > **Thank you for the additional experiments**
> >
> > Thank you for the additional experiments. This makes sense and overall I remain positive about this paper. I agree with the authors that saving an inverse DCT for downstream deployment is a good reason to absorb the DCT into the loss.

---

> ### Author Response · Authors · 2025-08-06
> **Further Response**
>
> Dear Reviewer,
>
> We sincerely appreciate your review and follow-up comment. We’re glad that our additional experiments addressed the concerns. Please don’t hesitate to reach out if there are any further questions. Thanks again for your valuable time and feedback !

---

### Official Review · Reviewer_nYM5 · 2025-07-03

**Clarity:** 3
**Significance:** 3
**Originality:** 2
**Rating:** 4
**Confidence:** 4

**Summary:**

FreqPolicy accelerates flow-based visuomotor policies by adding a frequency-domain consistency loss. Velocities at different flow timesteps are projected with a DCT and forced to share similar spectra, while an adaptive weighting scheme emphasizes the most mismatched bands. Combined with the standard flow-matching loss, this yields a one-step policy that matches or exceeds multi-step diffusion/flow baselines on various simulated and real-robot benchmarks.

**Questions:**

1. How much gain comes specifically from frequency projection versus plain time-domain consistency?
2. Can the loss be applied to diffusion policies in general?
3. Does the adaptive weighting ever over-focus noisy high-frequency band? how is stability ensured?
4. In the OpenVLA experiment, did you fine-tune or retrain the policy head from scratch?

**Ethical Concerns:**

["NO or VERY MINOR ethics concerns only"]

**Final Justification:**

The rebuttal has addressed most of my questions, and the discussions from other reviewers clearly suggest that the paper is above the acceptance threshold. Therefore, I will maintain my rating as weak accept.

**Limitations:**

The focus is on flow models; generality to other generators and combination with compression techniques are future work. Societal impact is minimal. Limitations are adequately acknowledged.

**Quality:**

3

**Strengths And Weaknesses:**

Strengths. The work is the first to exploit frequency consistency for robot policies, offering a fresh angle on the speed-versus-quality trade-off. Mathematics and implementation details are clear, and experiments are thorough. Simulation tasks plus real-world trials show the one-step model outperforming Consistency Policy, SDM, OneDP, and FlowPolicy while being 5–20× faster than diffusion. The method removes the need for a pretrained teacher and integrates cleanly into a large vision-language-action model, demonstrating practical relevance.

Weaknesses. The contribution, though clever, is incremental on top of existing consistency/flow frameworks; its novelty lies mainly in switching to the frequency domain. Applicability is demonstrated only for flow-matching; it is unclear whether diffusion or discrete-action policies would benefit. Ablations do not fully isolate how much improvement comes from the spectral projection itself, and hyper-parameter sensitivity (e.g., DCT length, loss weighting) is not discussed.

---

> ### Author Rebuttal · Authors · 2025-07-31
>
> We appreciate your review of our paper. Please find our responses to the concerns below:
>
> ------
> **Weakness:** *The contribution, though clever, is incremental on top of existing consistency/flow frameworks; its novelty lies mainly in switching to the frequency domain.*
>
> **A:** In essence, FreqPolicy remains a flow-based approach for robotic visuomotor policies and is one of a series of methods that leverage consistency principles to accelerate inference. We would like to point out that existing generative-based methods directly leverage existing generative models, such as diffusion and flow-matching from image generation fields, without considering the temporal characteristics of robotic manipulation. Moreover, we also reveal that in robotic manipulation, the distribution of frequency components within each action chunk varies over time throughout task execution. This variability arises because robotic manipulation sequences typically alternate between stationary and non-stationary motion phases. We propose an adaptive weighting scheme to capture this structured temporal variation.
>
> We believe that our findings offer the community a new perspective on the temporal characteristics of action generation.
>
>
>
> ------
> **Weakness & Q2:** *The applicable of FreqPolicy for diffuion-based polices.*
>
> **A:** Thanks for this valuable question. FreqPolicy was originally developed for flow-based visuomotor policies, using frequency consistency constraints to achieve stronger one-step inference performance. In principle, the same frequency-domain consistency idea can be applied to diffusion-based visuomotor policies, with an appropriately adapted formulation.
>
> To test its feasibility, we integrated frequency-domain consistency into a diffusion-based policy. Specifically, we built on the Consistency Policy [1], which leverages the classic diffusion model EDM [2]. We enhanced the second training process of Consistency Policy by enforcing a frequency-domain consistency constraint: the spectral representations of transitions from any two timesteps to a common third one must match.
>
> Thus, we compare two settings:
>
> A. Original Consistency Policy (a diffusion-based policy work).
>
> B. Consistency Policy with Freqpolicy adaptation, as detailed above.
>
> Concretely, we conducted experiments using the RoboMimic benchmark—leveraging the standard experimental protocol established by Consistency Policy. We run each variant across three random seeds and report the average results below:
>
> | Variant                             | Lift |    Can    |   Square  | Transport |  ToolHang |
> | ----------------------------------- | :--: | :-------: | :-------: | :-------: | :-------: |
> | A | 1.00 | 0.98±0.01 | 0.92±0.02 | 0.84±0.04 | 0.71±0.03 |
> | B | 1.00 | 0.99±0.01 | 0.91±0.02 | 0.87±0.03 | 0.75±0.03 |
>
> These results demonstrate that adding frequency-domain consistency to Consistency Policy yields clear improvements over the original time-domain loss, particularly on harder tasks (Transport: +0.03; ToolHang: +0.04). This confirms that our frequency-domain paradigm effectively extracts structured temporal information in diffusion-based visuomotor policies.
>
>
> ------
> **Weakness & Q1:** *Gain comes specifically from frequency projection versus plain time-domain consistency.*
>
> **A:** Sorry for the confusion. In our paper’s experiments, we demonstrated that frequency-domain consistency alone yields performance improvements over time-domain consistency.
>
> In Table 4, we conducted ablation studies to validate the effect of frequency-domain consistency, and the relevant data are as follows:
>
> | Variant  | Can  | Square | Transport | ToolHang |
> | -------- | ---- | ------ | --------- | -------- |
> | Model #2 | 0.98 | 0.92   | 0.84      | 0.76     |
> | Model #5 | 0.97 | 0.93   | 0.92      | 0.88     |
>
> Model #2 was trained using only standard time-domain consistency, while Model #5 employed a fixed frequency-domain consistency loss without adaptive weighting. Model #5 outperforms Model #2 on the harder tasks, demonstrating that this gain comes solely from frequency projection.
>
> To further validate this result, we evaluated both models on all 11 tasks from the Hard and Very Hard splits of MetaWorld, using three independent seeds per task and reporting the mean success rates below.
>
> |                 | Model #2      | Model #5      |
> | --------------- | ------------- | ------------- |
> | Avg. Results | 0.5935±0.0422 | 0.6101±0.0454 |
>
>
> Indeed, this comparison is crucial to reveal the importance of frequency transformation. We will further emphasize these comparisons in the revision.
>
>
>
> ------
> **Weakness:** *hyper-parameter sensitivity, e.g., DCT length, loss weighting, etc.*
>
> **A**: For the DCT transform, we apply a DCT-II to the time-series signal of each action dimension, whose length equals the action chunk (denoted by H in the paper). By default, this produces frequency-domain coefficients of the same length. To account for the influence of all frequency components, we do not filter or truncate any coefficients.  Instead, we use the full set of coefficients to compute the losses in Equations 12 and 14. This is a potential investigation into whether the length of DCT can be compressed; we will further study it.
>
> Regarding loss weighting, we also performed ablation studies on the key hyperparameter $\lambda$. Assuming our learning objective $\mathcal{L}\_{\text{total}} = \mathcal{L}\_{\text{freq}} + \lambda\ \mathcal{L}\_{\text{fm}}$, we set $\lambda=1$ in the results reported in our original paper. We then ran a series of experiments varying $\lambda$, and the detailed results are shown below:
>
> |                 | $\lambda=0.03$ | $\lambda=0.1$ | $\lambda=0.3$ | $\lambda=0.6$ | $\lambda=1$   | $\lambda=1.2$ | $\lambda=10$ |
> | :-------------: | :------------- | ------------- | ------------- | ------------- | ------------- | ------------- | ------------ |
> | Avg. Results | 0.5915±0.0348  | 0.6013±0.0326 | 0.6110±0.0357 | 0.617±0.0301  | 0.6148±0.0382 | 0.6091±0.0381 | 0.6148±0.044 |
>
> These experiments were conducted on 11 tasks from the Hard and Very Hard splits of MetaWorld, using three random seeds per task and reporting the average performance. We found that, for $\lambda$ within a typical range, performance fluctuates only marginally.
>
> We will include these details in the revision to address concerns about hyperparameter sensitivity.
>
>
>
> ------
> **Q3:** *Does adaptive weighting ever over-focus noisy high-frequency bands? how is stability ensured?*
>
> **A:** We are not sure whether the reviewer's mention of "noisy high-frequency bands" refers to 1) early-stage noise during generation or 2) high-frequency noise inherent in the training data.
>
> For case 1:
> The goal of our adaptive weighting is to accelerate convergence during training by emphasizing frequency components with greater discrepancies. Early high-frequency noise is amplified in the initial stages of training through our method, but this noise gradually diminishes as training progresses, with its weighting decreasing as the discrepancies lessen. As a result, the discrepancy reduces over time, and the weighting for these components decreases, ensuring stability.
>
> For case 2:
> If the training data contains inherent high-frequency noise, our method may be affected. Excessive noise can indeed impact the final performance. However, I believe that such noise will influence all imitation learning methods to some extent. We will acknowledge this limitation in the paper and consider potential strategies to mitigate such noise in future work, possibly through filtering techniques.
>
>
> ------
> **Q4:** *In the OpenVLA experiment, did you fine-tune or retrain the policy head from scratch?*
>
> **A:** We fine-tuned the pre‑trained OpenVLA‑7B checkpoint rather than training the policy head from scratch. Although OpenVLA’s original large‑scale pre‑training uses an autoregressive detokenization objective, OpenVLA‑OFT has demonstrated that one can successfully adapt its weights for simple regression and diffusion heads. Following this precedent, we directly fine‑tuned OpenVLA‑7B for both flow matching and FreqPolicy, and our experimental results confirm that this approach is effective. We will clarify these details more explicitly in the revision.
>
>
>
> [1] Aaditya Prasad, Kevin Lin, Jimmy Wu, Linqi Zhou, and Jeannette Bohg. Consistency policy: Accelerated visuomotor policies via consistency distillation. In Robotics: Science and Systems, 2024.

---

> ### Author Response · Authors · 2025-08-06
> **Further Response**
>
> Dear Reviewer,
>
> We sincerely thank you for your careful review of our submission and for acknowledging our responses. We truly appreciate the time and thought you’ve dedicated to engaging with our work.
>
> To address your concerns, we conducted additional ablation studies to isolate the role of frequency projection, and we further validated the feasibility of frequency-domain consistency within diffusion policies. If you have any further questions or would like additional clarification at any point, please don’t hesitate to reach out — we would be more than happy to provide further details. If our responses have resolved your concerns, we would be deeply grateful for your consideration of a higher rating, which would serve as strong encouragement for our work.
>
> Once again, thank you for your valuable feedback and thoughtful consideration.

---

### Official Review · Reviewer_kyT9 · 2025-07-03

**Clarity:** 2
**Significance:** 3
**Originality:** 2
**Rating:** 5
**Confidence:** 3

**Summary:**

The authors investigate methods for improving generative policies in robotic control. The problem they identify is the use of iterative generative models when generating actions, which reduce action sampling rates. They suggest that a solution is to introduce temporal consistency loss functions as a means of enhancing single-step prediction policies. The authors apply this loss function to a flow-matching policy and demonstrate the efficacy of this approach through extensive simulation and real-world experiments, showing improvements in robot control while increasing the action frequency rate.

**Questions:**

- Related Work line 95: How does the author’s work differ from the various previous works on generative models in visuomotor policies?
- Line 176 - Isn’t this technically preliminary work if you’re using the same loss as other papers?
- Equation 8 - Should the target be a_1 - a_t? As written, my understanding is that v_\theta should always predict the whole duration from a_0 to a_1 regardless of the time or a_t value given.
- Equation 10 line 204:  If r <s < u, wouldn’t it be better to explicitly show this in the notation when sampling? I.e. r \sim U(0, 1), s \sim U(r, 1), u \sim U(s, 1)? Or something similar?
- Equation 11, 13, what is “H” here?
- What research questions are the experiments addressing?
- Table 1: are the reported improvements statistically significant by an appropriate hypothesis test?
 - How many runs were used in the ablation experiments shown in Table 4? It seems strange there aren’t standard deviations reported.

**Ethical Concerns:**

["NO or VERY MINOR ethics concerns only"]

**Final Justification:**

I am maintaining my score, which was to accept the paper. Many of my concerns were minor and the author's have addressed them in the rebuttal process.

**Limitations:**

No
The limitations section (lines 335 - 340) doesn’t list any real limitations: “Our work focuses on accelerating flow matching through frequency domain modelling.” which is followed by a vague comment that the authors might or might not try to enforce other things on temporal motions. The authors should consider the potential problems or pitfalls of using their proposed loss function more deeply. An obvious limitation in our assessment is that the performance gains of the full loss function may depend on task complexity (e.g., Table 4 - Can or square results show minimal gains, whereas Transport and Toolhang show an 8% and 12% improvement in success rate, respectively). As we mention above, if only a single model is trained and evaluated on, this is definitely an important limitation.

**Paper Formatting Concerns:**

- the abstract is a bit long
- Many acronyms used but not explained (e.g. EDM model line 44, SDM line 47)
- 40 - 53: this paragraph could probably be compressed or have content moved to related work if the details help distinguish the author’s work from prior research
- The whole introduction is too long; the authors could probably cut lines 68 - 83 without much loss of understanding to readers
- Related Work: it’s too detailed and covers a lot of ground. It might be good to consider ways of summarizing works more succinctly. Do readers need to know every individual policy proposed in prior research to appreciate the author’s work?
- Section 3: This is a personal preference, but having a section preamble can be beneficial in setting the stage for what will be discussed in Section 3. As part of this,  3.2 could be moved to establish the assumptions the authors are making throughout each section, saving space on having a header for “Task Formulation.”
- Equation 10 might be better to explicitly state the specific similarity function used instead of keeping it abstract
- Figure 3: It took a bit of time to parse the connection that, for prior models to match success rates to the author’s work, they would lead to the robot moving much slower. Perhaps group success rates & Frequency next to each other by task?
- Figure 3: Get rid of the underscore (e.g. “task_1”). It does not look good.

**Quality:**

3

**Strengths And Weaknesses:**

Strengths:

- The authors' proposed method seems to address issues of iterative sampling from prior generative policies, showing similar or better success rates while requiring fewer model evaluations to generate action sequences
- The evaluation is extensive, utilizing 53 tasks across three simulation benchmarks to assess the efficacy of this platform.
- Real robot experiments are performed, which is always an important demonstration for robotic-focused applications.

Weaknesses

Our biggest concern is distinguishing the author’s contribution from prior works. The authors can address this issue by refining their writing, which we found to be verbose in certain sections. We discuss suggestions on this in the “Paper Formatting” section.

 Some of the reported results could benefit from additional analysis, such as performing appropriate hypothesis testing to verify that any improvements shown on average are also statistically significant. It is surprising that standard deviations are excluded in some sections for evaluation, suggesting that the authors did not perform multiple runs of the experiments during model training. If the author’s experiments report results from a single trained model for each testing scenario rather than several models trained with different seeds, this is a weakness of the paper. In this case, the authors should clearly state this as a limitation of their results. We suspect it’s unlikely, as this is behavior cloning, but perhaps different policies would lead to measurably different performance results. Additionally, as we stated, this makes it hard to perform a statistical assessment of their model.

---

> ### Author Rebuttal · Authors · 2025-07-31
>
> We sincerely thank you for the thorough review of our work, as well as for the constructive suggestions on the paper formatting. Please find our responses below:
>
> ------
> **Weakness & Q1:** *Difference between FreqPolicy and various previous works on generative models in visuomotor policies.*
>
> **A:** Our work is built upon the existing flow-based policy workflow. However, existing generative models in visuomotor policies directly utilize generative models from the image generation field, without considering the temporal characteristics of robotic manipulation. In our work, we leverage the temporal information to improve the action generation for a flow-based policy. Moreover, we also reveal that in robotic manipulation, the distribution of frequency components within each action chunk varies over time throughout task execution. This variability arises because robotic manipulation sequences typically alternate between stationary and non-stationary motion phases. We propose an adaptive weighting scheme to capture this structured temporal variation.
>
> We will emphasize the difference in our revision.
>
>
> ------
> **Q2:** *Line 176 - Isn’t this technically preliminary work if you’re using the same loss as other papers?*
>
> **A:** Yes, it is a common objective used in the prior flow‑based approaches, which we cite in Line 178. In FreqPolicy, we adopt it as the base objective and then impose our frequency consistency constraint based on it. For rigor, we have retained the full formulation of this loss in the paper.
>
>
> ------
> **Q3:** *Equation 8 - Should the target be a_1 - a_t?*
>
> **A:** The velocity network $v_\theta$ is meant to match the constant velocity over the interval [0,1], so the target is $a_1−a_0$. This basic formulation is consistent with prior flow‑matching works (e.g., PI0 [1], GR00T-N1 [2]).  We will detail the description of Eq.8 in the revision.
>
>
>
> ------
> **Q4:** *Equation 10 line 204: If r <s < u, wouldn’t it be better to explicitly show this in the notation when sampling? I.e. r \sim U(0, 1), s \sim U(r, 1), u \sim U(s, 1)? Or something similar?*
>
> **A:** Good suggestion. In the revision, we will explicitly specify the sampling range for each temporal variable to clarify their interrelationships.
>
>
> ------
> **Q5:** *Equation 11, 13, what is “H” here?*
>
> **A:** As noted at Line 160, $H$ denotes the action chunk length, e.g., the number of timesteps for each action dimension.
>
>
> ------
> **Q6:** *What research questions are the experiments addressing?*
>
> **A:** The experiments were mainly to validate the effectiveness of our method, which shows that the temporal information is crucial in robotic manipulation tasks. And it outperforms existing works that directly implement generative models from image generation without considering the temporal characteristics of robotic manipulation.
>
> Speccifically, Our experimental suite is designed to answer 3 questions: (1) whether enforcing consistency in the frequency improves performance over existing multi‑step diffusion and flow‑matching baselines (as shown in Tables 1); 2) whether our frequency consistency loss can be fine‑tuned on large pre‑trained VLA models to yield tangible gains (Table 3); 3) what the individual contributions of fixed frequency consistency and the adaptive weighting are relative to a vanilla one‑step flow‑matching objective (Table 4).
>
>
> ------
> **Weakness & Q7:** *Table 1: are the reported improvements statistically significant by an appropriate hypothesis test?*
>
> **A:** Yes. To ensure statistical validity, we followed the protocol of Consistency Policy [3] and evaluated on the RoboMimic benchmark via 3 random seeds. We then report the mean and standard deviation metric for each task.
>
>
> ------
> **Q8:** *How many runs were used in the ablation experiments shown in Table 4?*
>
> **A:** We apologize for the omission. For the ablation experiments in Table 4, we ran each variant with three independent random seeds and reported the mean success rate. We will include the corresponding standard deviations in the revision.
>
>
> ------
> **Q9:** *No The limitations section (lines 335 - 340) doesn’t list any real limitations.*
>
> **A:** Thanks for the constructive advice. Indeed, regarding the task complexity, in RoboMimic, the Can, Lift, and Square tasks are relatively simple, so most existing methods show similar performance on these simple tasks, whereas ToolHang and Transport are more challenging and offer greater room for improvement. We will explicitly state in the revision that our ablation studies should be conducted across a broader range of benchmarks and clarify the performance gains observed for tasks of varying difficulty.
>
> We will also consider the potential problems and include them in our limitations, such as our work potentially can be used for diffusion-based policy as well, yet we do not comprehensively study this in this work. And as our work is based on the consistency constraints principle, it consumes more memory during training since it requires twice the forward for two random samples.
>
>
> ------
> **Q10:** *Paper Formatting*
>
> **A:**  We will address all formatting suggestions as follows:
>
> 1. Shorten the abstract to approximately 20 lines.
> 2. Remove any undefined acronyms from the abstract and verify that all acronyms are defined at first use throughout the paper.
> 3. Compress Lines 40–53 to retain only the content directly relevant to introducing FreqPolicy.
> 4. Remove the redundant sentences in Lines 68–83 of the Method section.
> 5. Prune the Related Work to focus on the most pertinent prior studies, eliminating less directly relevant references.
> 6. Remove the “3.2 Task Formulation” and “3.3.1 Training Strategy” headings and either decouple or reformat the Task Formulation content for clarity.
> 7. Revise Equation 10 to improve its readability.
> 8. Redesign Figure 3’s bar plots to center each task for clearer comparison.
> 9. Rename “task_1” to “task 1” for improved presentation.
>
> Thanks again for this careful and comprehensive review. We will incorporate these changes in the revision.
>
>
>
> [1] Kevin Black, Noah Brown, Danny Driess, Adnan Esmail, Michael Equi, Chelsea Finn, Niccolo
> Fusai, Lachy Groom, Karol Hausman, Brian Ichter, et al. pi0: A vision-language-action flow
> model for general robot control. arXiv preprint arXiv:2410.24164, 2024.
>
> [2] Johan Bjorck, Fernando Castañeda, Nikita Cherniadev, Xingye Da, Runyu Ding, Linxi Fan,
> Yu Fang, Dieter Fox, Fengyuan Hu, Spencer Huang, et al. Gr00t n1: An open foundation model
> for generalist humanoid robots. arXiv preprint arXiv:2503.14734, 2025.
>
> [3] Aaditya Prasad, Kevin Lin, Jimmy Wu, Linqi Zhou, and Jeannette Bohg. Consistency policy: Accelerated visuomotor policies via consistency distillation. In Robotics: Science and Systems, 2024.

---

> > ### Comment · Reviewer_kyT9 · 2025-08-04
> > **Comments were addressed, just a few things to clarify.**
> >
> > Q6: What research questions are the experiments addressing?
> > - The format of the questions listed were clear and should probably be incorporated directly into the writing of the section.
> >
> >
> > Weakness & Q7: Table 1: are the reported improvements statistically significant by an appropriate hypothesis test?
> > - We were referring to running something like a T-test or other such appropriate test and reporting for example the p-value of results. If this was not done, we do consider this weakness of the paper, and prior research frankly if they did not conduct similar tests.

---

> ### Author Response · Authors · 2025-08-06
> **Response to Q6 and Q7**
>
> Dear Reviewer,
>
> We sincerely appreciate your reading of our response and your further clarification. Please find our brief follow-up reply below:
>
> ------
>
> **Regarding Q6:**
>
> We will incorporate the research questions explicitly at the beginning of the experimental section in the revised version for clarity.
>
> ------
>
> **Regarding Weakness & Q7:**
>
> We acknowledge that we followed the common practice in prior work of reporting comparisons using mean and standard deviation, which we agree is a limitation in terms of statistical rigor in this field.
>
> We believe that applying hypothesis testing, such as a t-test, is a more scientific and accurate approach. Due to time constraints, we made a trade-off between training time and task complexity, and thus conducted the additional analysis on the Robomimic **transport** task. We ran 6 trials with different seeds and performed a paired t-test using scipy.stats.ttest_rel. We compare **FreqPolicy** against the classic **Consistency Policy** and our re-implementation of **Consistent-FM**.
>
> Specifically, we report in the table below the success rates of the three methods across six random seeds.
>
> |                    | trial 1 | trial 2 | trial 3 | trial 4 | trial 5 | trial 6 |
> | ------------------ | :-----: | :-----: | :-----: | :-----: | :-----: | :-----: |
> | Consistency Policy |  0.80   |  0.82   |  0.75   |  0.80   |  0.78   |  0.83   |
> | Consistent-FM      |  0.83   |  0.78   |  0.80   |  0.75   |  0.82   |  0.80   |
> | FreqPolicy         |  0.92   |  0.88   |  0.92   |  0.90   |  0.90   |  0.88   |
>
> Based on these results, we further conduct paired *t*-tests to evaluate statistical significance. For clarity and direct hypothesis testing, we convert the original two-sided $p$-values into one-sided $p$-values, in order to test the hypothesis *"our method outperforms the baseline"*.
>
> |            | Consistency Policy | Consistent-FM |
> | ---------- | :----------------: | :-----------: |
> | FreqPolicy |       0.0001       |    0.0013     |
>
> As the table indicates, the improvement achieved by our method is statistically significant ($p < 0.05$), providing evidence for the effectiveness of our approach. We will include additional hypothesis testing under more task settings in the revision to enhance the scientific rigor of our method comparisons.
>
> Thank you once again for your insightful feedback and kind consideration.

---

### Note · Authors · 2025-08-15

Dear ACs, SACs, and Reviewers,

We sincerely appreciate your time, effort, and constructive feedback for our work.

Across the reviews, our work has been widely recognized as **well-motivated** (`fS2J`, `4VTk`), **novel** (`4VTk`, `nYM5`), **experimentally solid**(`kyT9`, `nYM5`, `fS2J`, `4VTk`), and **clearly written** (`nYM5`, `4VTk`). Following the rebuttal and discussions, all reviewers maintained positive scores.

The main concerns raised in reviews include:
- Whether experiments are repeated and statistically significant (`kyT9`);
- Standalone benefit of frequency-domain projection (`nYM5`);
- Applicability to diffusion-based policy (`nYM5`);
- Comparison with a similar implementation (`fS2J`);
- Robustness to hyperparameters and frequency projections (`4VTk`).

In the rebuttal, we further addressed these concerns with additional experiments and analyses:
- Conduct extra experiments on the Robomimic transport task, with t-tests confirming statistically significant improvements over key baselines (`kyT9`);
- Add detailed ablations comparing the time- and frequency-domain consistency, verifying the effectiveness of the latter (`nYM5`);
- Integrate frequency-domain consistency into a classic diffusion policy, achieving performance gains (`nYM5`);
- Compare with the similar approach suggested by `fS2J`, showing comparable performance but higher theoretical inference efficiency (`fS2J`);
- Test different hyperparameter settings and projection methods, confirming robustness (`4VTk`).

In summary, as highlighted by the reviewers, FreqPolicy contributes to the community by:
- **Novelty**: the first to employ frequency consistency to capture temporal knowledge between action chunks (`4VTk`, `nYM5`), improving manipulation performance.
- **Generalization**: applicable to both flow matching and diffusion policies (`nYM5`), it enhances one-step action generation and can be integrated into existing sampling-based VLAs to boost inference speed (`4VTk`).
- **Impact**: valuable to both academia and industry: it offers a new angle for multi-modal one-step action generation (`nYM5`) and improves real-robot deployment via faster inference (`4VTk`).
- **Effectiveness**: extensive simulation and real-robot experiments validate its efficacy (`kyT9`, `nYM5`, `fS2J`, `4VTk`), while visual analyses reveal action frequency diversity, offering new insights for visuomotor policy design.

Once again, we thank ACs, SACs, and Reviewers for their efforts in reviewing our paper.

---

### Decision · Program_Chairs · 2025-09-17

**Decision:**

Accept (poster)

**Comment:**

This paper investigate methods to improve generative policies for robotic control. Iterative generative models in action generation reduce sampling rates, and then the author propose temporal consistency loss to enhance single-step action prediction policies. Applying this loss to a flow-matching policy, they validate it via extensive simulation and real-world experiments.

All reviewers vote for acceptance , and the AC accordingly recommends accepting this submission.